# On the role of climate modes in modulating the air-sea CO$_2$ fluxes in Eastern Boundary Upwelling Systems

Riley X. Brady[1], Nicole S. Lovenduski[1], Michael A. Alexander[2], Michael Jacox[2,3], and Nicolas Gruber[4]

[1]Department of Atmospheric and Oceanic Sciences and Institute of Arctic and Alpine Research, University of Colorado, Boulder, CO, USA
[2]NOAA/ESRL, Boulder, CO, USA
[3]NOAA/SWFSC, Monterey, CA, USA
[4]Environmental Physics, Institute of Biogeochemistry and Pollutant Dynamics, ETH Zürich, Zürich, Switzerland

**Correspondence:** Riley X. Brady (riley.brady@colorado.edu)

**Abstract.** The air-sea CO$_2$ fluxes in Eastern Boundary Upwelling Systems (EBUS) vary strongly in time and space with some of the highest flux densities globally. The processes controlling this variability have not yet been investigated consistently across all four major EBUS, i.e., the California (CalCS), Humboldt (HumCS), Canary (CanCS), and Benguela (BenCS) Current Systems. In this study, we diagnose the climatic modes of the air-sea CO$_2$ flux variability in these regions between 1920-2015, using simulation results from the Community Earth System Model Large Ensemble (CESM-LENS), a global coupled climate model ensemble that is forced by historical and RCP8.5 radiative forcing. Differences between simulations can be attributed entirely to internal (unforced) climate variability, whose contribution can be diagnosed by subtracting the ensemble mean from each simulation. We find that in the CalCS and CanCS, the resulting anomalous CO$_2$ fluxes are strongly affected by large-scale extratropical modes of variability, i.e., the North Pacific Gyre Oscillation (NPGO) and the North Atlantic Oscillation (NAO), respectively. The CalCS (CanCS) has anomalous uptake (outgassing) of CO$_2$ during the positive phase of the NPGO (NAO). In contrast, the HumCS is mainly affected by El Niño Southern Oscillation (ENSO), with anomalous uptake of CO$_2$ during an El Niño event. Variations in dissolved inorganic carbon (DIC) and sea surface temperature (SST) are the major contributors to these anomalous CO$_2$ fluxes, and are generally driven by changes to large-scale gyre circulation, upwelling, the mixed layer depth, and biological processes. A better understanding of the sensitivity of EBUS CO$_2$ fluxes to modes of climate variability are key to improve our ability to predict the future evolution of the atmospheric CO$_2$ source/sink characteristics of the four EBUS.

## 1 Introduction

The four major Eastern Boundary Upwelling Systems (EBUS) occur at the eastern edges of subtropical gyres in the Atlantic and Pacific oceans – the California (CalCS), Humboldt (HumCS), Canary (CanCS), and Benguela (BenCS) Current Systems. These regions are characterized by seasonal or permanent equatorward winds that cause upwelling due to both offshore Ekman transport as well as wind stress curl-driven Ekman suction within the first 200 km of the coastline (Chavez and Messié, 2009). Upwelling delivers waters rich in nutrients to the surface, fueling primary production and ultimately supporting fisheries that

are highly productive relative to the small surface area they cover (Ryther, 1969). Upwelled waters also have a high dissolved inorganic carbon (DIC) content, which causes initially at the surface an elevated partial pressure of carbon dioxide ($p$CO$_2$), a low pH, and a low carbonate ion concentration, i.e., a low saturation state with regard to CaCO$_3$ minerals. The high productivity fueled by the upwelling pushes the surface ocean pCO$_2$ quickly down again, and also raises the pH and the carbonate ion concentration again (Turi et al., 2014). Further processes including entrainment of subsurface waters, horizontal advection and mixing, vertical mixing, temperature changes, respiration, and calcium carbonate formation and dissolution (DeGrandpre et al., 1998; King et al., 2007) affect the surface pCO$_2$. Thus, the resulting pCO$_2$ distribution is the result of a complex interplay of physical and biological processes. These terms combine to dictate oceanic $p$CO$_2$, which drives the $p$CO$_2$ gradient between the ocean and atmosphere ($\Delta p$CO$_2$), thus determining the sign and magnitude of the air-sea CO$_2$ flux.

Although coastal oceans around the world have a small net contribution to the global air-sea CO$_2$ flux, they are characterized by a high CO$_2$ flux density, or magnitude of air-sea CO$_2$ exchange per unit area (Laruelle et al., 2010, 2014; Gruber, 2015; Laruelle et al., 2017). Low-latitude upwelling systems, such as the HumCS and CanCS, tend to be net outgassing systems, due to their relatively warm waters and persistent upwelling, which are not fully compensated for by enhanced biological productivity. Because of their colder temperatures and greater biological production, mid-latitude systems, such as the CalCS and BenCS, act as weak CO$_2$ sinks that can become CO$_2$ sources during certain seasons (Borges and Frankignoulle, 2002; Hales et al., 2005; Cai et al., 2006; Gregor and Monteiro, 2013). Surface ocean $p$CO$_2$ and thus air-sea CO$_2$ flux in EBUS exhibits high temporal variability at sub-seasonal, seasonal, and interannual time scales (Friederich et al., 2002; González-Dávila et al., 2009; Leinweber et al., 2009; Evans et al., 2011; Turi et al., 2014), driven by different sets of processes: Fundamentally, one can differentiate between variability arising from the processes that are purely internal to the climate system, and those that represent "external forcings", i.e., processes that impact the climate system from outside. The latter external processes can be further separated into natural and anthropogenic. The former includes variations induced by e.g., volcanic eruptions or changes in solar activity, while the latter includes changes in the concentration of greenhouse gases and other radiatively active constituents, or human-made changes in albedo. The internal variability can arise from within a subsystem itself (e.g., baroclinic instabilities leading to the formation of mesoscale eddies), or from the unforced interaction between components of the climate system.

So far, relatively few studies have truly assessed the longer-term variability of the air-sea CO$_2$ fluxes in EBUS, regardless of whether these variations are internal or forced. Notable exceptions are Friederich et al. (2002) who analyzed the impact of the El Niño Southern Oscillation (ENSO) in the CalCS and Chavez et al. (1999) who did the same for the HumCS. El Niño (La Niña) tends to cause uptake (outgassing) anomalies in these systems, primarily through modifications to the thermocline depth and upwelling rates of nutrient- and carbon-rich waters, which in turn alters biological activity. However, upwelling-favorable winds can persist during some El Niños in the HumCS (Huyer et al., 1987), leading to persistent or enhanced outgassing of CO$_2$ to the atmosphere (Torres et al., 2003). Longer-term fluctuations in the CalCS arise from Pacific Decadal Variability. Although studies have linked the Pacific Decadal Oscillation (PDO) and North Pacific Gyre Oscillation (NPGO) to low frequency changes in upwelling rates, nutrient fluxes, ocean acidification, and fisheries in the CalCS (e.g., Chenillat et al., 2012; Turi et al., 2016; Chhak and Di Lorenzo, 2007; Di Lorenzo et al., 2008; Mantua et al., 1997), no work has been done to directly investigate

the effect of decadal variability on Pacific EBUS $CO_2$ fluxes in particular. However, studies have shown that a positive PDO intensifies the trade winds along the equatorial Pacific, leading to intensified upwelling and thus outgassing (Feely et al., 2006; Takahashi et al., 2003) and lower pH and saturation states (Turi et al., 2016). The response of HumCS $CO_2$ fluxes to the PDO might be similar. To the best of our knowledge, there have been no studies exploring $CO_2$ flux sensitivity to large modes of

climate variability in the two Atlantic EBUS. However, Cropper et al. (2014) found that the North Atlantic Oscillation (NAO) plays a major role in modulating interannual variability of coastal upwelling in the CanCS and Borges et al. (2003) link the NAO to decadal variability in sardine catch. Variability in upwelling and biology in the BenCS has been linked to Benguela Niños (Shannon et al., 1986), ENSO teleconnections, and the Southern Annular Mode (Reason et al., 2006; Hutchings et al., 2009). However, decadal-scale oscillations like the NAO or PDO do not appear to be present in the South Atlantic (Hutchings

et al., 2009). In summary, prior research has illuminated the large temporal variability of $p$CO$_2$ and $CO_2$ fluxes in EBUS and few have analyzed the impacts of single climate events on anomalous $CO_2$ fluxes in the CalCS and HumCS. Past studies tend to focus instead on linking internal climate variability to upwelling, nutrients, and fish catch. Our study aims to address this gap by identifying the major mode of climate variability associated with anomalous $CO_2$ fluxes in the major EBUS and by further investigating the dynamics that underpin these anomalies. We are thus particularly interested in elucidating the role of

internal climate variability and less on the forced (external) sources of variability.

Direct observation is of course the most desirable tool for understanding the real world, but it is not feasible for this study due to the sparsity of $p$CO$_2$ and air-sea $CO_2$ flux measurements and the relatively short length of observational time series. Regional hindcast simulations are beneficial for their higher spatial resolution and more accurate representation of a specific EBUS's dynamics, but they are limited to the analysis of a single EBUS, preventing a synchronous view across EBUS with a consistent

modeling tool. Further, single realizations (i.e., observational products and hindcast simulations) provide a limited sample size of internal variability and confound the impacts of external forcing with internal climate variability. The limited sample size is problematic, because internal stochastic noise causes ENSO events to evolve differently in the tropics (Capotondi et al., 2015). Moreover, mid-latitude atmospheric noise can obscure the tropical-extratropical connections associated with climate modes such as ENSO, causing a diversity of responses in EBUS (Deser et al., 2017, 2018). The latter makes it difficult to isolate the

internal variability in $CO_2$ fluxes from the seasonal cycle and other external forcing. One solution to this problem is to use a single-model ensemble that is derived by introducing perturbations to the initial state of the climate system. This gives rise to a set of realizations with unique representations of internal climate variability and gives one access to many hundred ENSO events. By performing historical experiments with increasing atmospheric $CO_2$ rather than a long control simulation, we can account for variability in air-sea flux of both natural $CO_2$ (the component of ocean $CO_2$ in equilibrium with pre-industrial

atmospheric $CO_2$) and anthropogenic $CO_2$.

In this study, we utilize output from the single-model Community Earth System Model "Large Ensemble" (CESM-LENS; Kay et al., 2015; Lovenduski et al., 2016) to identify major modes of climate variability that are associated with air-sea $CO_2$ fluxes in the major EBUS. To better identify these internal variability-driven fluxes, we subtract from each ensemble member the ensemble mean fluxes, as the latter represents essentially just the externally forced component. This results in what we refer

to as the "anomalous" fluxes. The availability of 34 simulations allows us to find statistically robust relationships between these

resulting anomalous $CO_2$ fluxes and internal climate variability. We expand on this by investigating the physical and biological drivers that underpin these air-sea $CO_2$ flux anomalies.

## 2 Methods

### 2.1 Model Configuration

We utilize monthly mean output from 34 members of the CESM-LENS (Kay et al., 2015; Lovenduski et al., 2016), which is derived from the Community Earth System Model, version 1, with the Community Atmosphere Model, version 5 (CESM1(CAM5); Hurrell et al., 2013). Along with standard atmosphere, ocean, land, and sea ice components, the CESM-LENS simulations include land and ocean biogeochemistry. The ocean biogeochemical component of CESM1(CAM5) is the Biogeochemical Elemental Cycling (BEC) model, which has three phytoplankton functional groups and tracks the cycling of C, N, P, Fe, Si, and O in the ocean. Further information on the implementation of BEC in CESM1 can be found in Moore et al. (2013) and Lindsay et al. (2014). The ocean component is the Parallel Ocean Program, version 2 (Smith et al., 2010) and has a nominal $1^o$ horizontal resolution with vertical resolution of 10 m through the upper 250 m, thereby resolving the Ekman layer. Due to the coarse horizontal resolution, neither curl-driven nor coastal upwelling is directly resolved, but both are represented in the model. A more detailed discussion of coastal upwelling in the CESM-LENS for the CalCS in particular can be found in Brady et al. (2017).

The 34 ensemble members of CESM-LENS were generated using round-off level (order $10^{-14}$ K) perturbations in the initial atmospheric temperature (Kay et al., 2015). This generates an ensemble of simulations that diverge solely due to the influence of internal variability. Each ensemble member was forced with common external forcing: historical radiative forcing from 1920 to 2005 and RCP8.5 radiative forcing from 2006 to 2100. Anthropogenic $CO_2$ was explicitly separated from natural $CO_2$ by computing air-sea $CO_2$ fluxes relative to a constant 284.7 ppm pre-industrial atmosphere and subtracting it from $CO_2$ fluxes responding to the evolving atmosphere under historical and RCP8.5 radiative forcing.

### 2.2 Upwelling Regions and Anomalies

Upwelling regions in each EBUS span approximately the $10^o$ latitude of most active upwelling as defined by Chavez and Messié (2009); we shifted the CanCS upwelling domain north by $9^o$ to capture the more intense upwelling off the Western Sahara in CESM1 (Table 1). Following Turi et al. (2014), the EBUS upwelling regions span from the coastline to 800 km offshore, tracking the model coastline. The black outlines in Figure 1e–h display these regions. The CESM-LENS ensemble mean incorporates the seasonal cycle and any long-term anthropogenic trends for a given variable. We therefore removed the ensemble mean from each simulation to create a time series of anomalies produced solely by internal climate variability. Note that for air-sea $CO_2$ flux, these anomalies represent internal variability in the contemporary flux of $CO_2$, or the combined variability of natural and anthropogenic $CO_2$.

## 2.3 Climate Indices

Throughout this paper, we regress anomaly time series of variables from each EBUS onto climate indices. Climate index time series were available for each ensemble member for the PDO, NAO, Atlantic Multidecadal Oscillation (AMO), ENSO, and Atlantic Meridional Overturning Circulation (AMOC) through the Climate Variability Diagnostics Package (CVDP; Phillips et al., 2014). We computed the NPGO index for each simulation following Di Lorenzo and Mantua (2016). These indices are available through the CESM-LENS project page (NPGO). Note that the NAO, NPGO, and PDO indices referred to in this study are based on normalized principal component time series and are thus reported in standard deviation ($\sigma$) units.

## 2.4 Model Equations and Statistical Analyses

Air-sea $CO_2$ fluxes in CESM are computed following the parameterization of Wanninkhof (2014):

$$F = k(U) \cdot K_0 \cdot (pCO_2^o - pCO_2^a),$$  (1)

where $k$ represents the gas transfer velocity (dependent on the wind speed (U) squared), $K_0$ the solubility of $CO_2$ in seawater, and $pCO_2^o$ and $pCO_2^a$ the partial pressures of $CO_2$ in the surface ocean and atmosphere, respectively.

We use a linear Taylor expansion to quantify the relative contribution of each variable to the overall air-sea $CO_2$ flux anomaly in response to unforced variability following Lovenduski et al. (2007) and Turi et al. (2014),

$$\Delta F = \frac{\partial F}{\partial U} \Delta U + \frac{\partial F}{\partial pCO_2^{oc}} \Delta pCO_2^{oc},$$  (2)

where $\frac{\partial F}{\partial U}$ and $\frac{\partial F}{\partial pCO_2^{oc}}$ are determined from the model equations and mean values in each EBUS. $\Delta$'s represent the linear regression of the given variable's anomalies onto a climate index. In addition to the linearization of the response, which results in us neglecting potentially important cross-derivative terms, we neglected further several terms in Equation 2. In particular, we neglected the role of temperature and salinity for the solubility $K_0$ and the gas transfer velocity $k(U)$. This is justified since the impact of salinity is small and the temperature sensitivity of the product $k(U) \cdot K_0$ is very small, owing to a strong cancellation effect. Further we neglected the contribution of changes in atmospheric $CO_2$, $pCO_2^a$, primarily because these variations are at least 10 times smaller than those seen in surface ocean $pCO_2^{oc}$.

The contribution from $\Delta pCO_2^{oc}$ is further decomposed into the contribution of changes in DIC, Alk, SST, and salinity,

$$\Delta pCO_2^{oc} = \frac{\partial pCO_2^{oc}}{\partial DIC} \Delta DIC + \frac{\partial pCO_2^{oc}}{\partial Alk} \Delta Alk + \frac{\partial pCO_2^{oc}}{\partial T} \Delta T + \frac{\partial pCO_2^{oc}}{\partial S} \Delta S.$$  (3)

The changes in DIC and Alk induced by freshwater fluxes have a strongly opposing influence on pCO₂, thus it is often more instructive to separate this effect on DIC and Alk from the others, and to summarize all salinity related terms into a freshwater term (see Lovenduski et al. (2007) for details). The remaining, non-salinity driven changes in DIC and Alk are referred to then as the salinity-normalized DIC (sDIC) and Alk (sAlk),

$$\Delta F = \frac{\partial F}{\partial U} \Delta U + \frac{S}{S_0} \frac{\partial F}{\partial DIC} \Delta sDIC + \frac{S}{S_0} \frac{\partial F}{\partial Alk} \Delta sAlk + \frac{\partial F}{\partial fw} \Delta fw + \frac{\partial F}{\partial T} \Delta T + \frac{\partial F}{\partial S} \Delta S.$$  (4)

Due to the significance of sDIC anomaly contributions to the total $CO_2$ flux anomaly in EBUS, we approximate the mechanisms controlling sDIC anomalies following Lovenduski et al. (2007),

$$\frac{d(sDIC')}{dt} = J'_{circ} + J'_{bio} + J'_{ex} \tag{5}$$

where $J'_{circ}$, $J'_{bio}$, and $J'_{ex}$ represent the sources and sinks of sDIC$'$ from circulation, biology, and $CO_2$ flux anomalies integrated over the upper 100m, respectively. We refer the reader to Lovenduski et al. (2007, their Equation 4) for additional details on these terms and their Appendix B for the computation of $J'_{bio}$ in particular.

We use a Pearson product-moment correlation for all linear correlations performed in this study (e.g., between area-weighted $CO_2$ flux anomalies and climate indices for each EBUS). Our null hypothesis is that the two time series being compared are uncorrelated, following the Student's $t$-distribution with a significance level of $\alpha = 0.05$.

Autocorrelation is prevalent in climate indices such as the NPGO and ENSO (Di Lorenzo and Ohman, 2013), and our annual smoothing further enhances autocorrelation in CalCS and CanCS air-sea $CO_2$ fluxes (see Sections 3.3.1 and 3.3.3). To compensate for this autocorrelation, we replace the $t$-statistic sample size $N$ with an effective sample size $N_{eff}$, which quantifies the number of statistically independent measurements:

$$N_{eff} = N \left( \frac{1 - r_1 r_2}{1 + r_1 r_2} \right) \tag{6}$$

where $r_1$ and $r_2$ are the lag-1 autocorrelation coefficients of the two time series being correlated (Bretherton et al., 1999; Lovenduski and Gruber, 2005).

We use a one-sided Mann-Kendall test to assess significance in trends (e.g., the long-term diffusion of anthropogenic $CO_2$ into EBUS). Our null hypothesis is that the trend is not significantly different from zero, with $\alpha = 0.05$.

## 3 Results

### 3.1 Model Evaluation

CESM-LENS air-sea $CO_2$ flux climatologies and $p$CO$_2$ seasonal cycles were compared to the SOM-FFN (Self-Organizing Map-Feed Forward Network) product from Landschützer et al. (2017) along the four major EBUS outlined by Chavez and Messié (2009). The SOM-FFN is a $CO_2$ flux product based on numerous observational datasets and is made available at monthly resolution spanning 1982-2015 at $1^o$x$1^o$ global resolution. Extensive details on and validation of the procedure can be found in Landschützer et al. (2013) and Landschützer et al. (2016). The SOM-FFN does not include coastal estimates, which presumably would have stronger outgassing due to coastal and curl-driven upwelling of DIC-enriched waters. Another important caveat is that the average number of $p$CO$_2$ observations in EBUS informing the SOM-FFN (637, 119, 517, and 195 for the CalCS, HumCS, CanCS, and BenCS, respectively) is on the order of the Southern Ocean (536), a notably undersampled region (Figure 2e and f; Bakker et al., 2016). An extension of the SOM-FFN approach to the coastal ocean by Laruelle et al. (2017) demonstrated that the coastal pCO$_2$ values are not substantially different from the offshore, so that the conclusions drawn from the open ocean Landschützer product still hold.

### 3.1.1 Air-sea $CO_2$ Flux Climatology

The mean state of Pacific EBUS $CO_2$ fluxes is particularly well-modeled in CESM-LENS (Figure 1). The CESM-LENS captures the meridional gradient of poleward uptake and equatorward outgassing of $CO_2$ in the CalCS (Figure 1e). In the HumCS, the model depicts the strong outgassing that is characteristic of a tropical upwelling system (Figure 1f). The $CO_2$ flux climatology in the Atlantic systems is more biased in the CESM-LENS, with a tendency for spurious or stronger outgassing than is suggested by the observational product. While the SOM-FFN portrays a meridional gradient of relatively weak $CO_2$ fluxes in the CanCS, the CESM-LENS simulates strong outgassing along the Western Sahara (Figure 1c and g). The BenCS has the most biased $CO_2$ flux climatology of the major EBUS in CESM-LENS. Although it simulates the proper meridional gradient, the ougassing cell is nearly $10^o$ too far south and is significantly stronger than in the SOM-FFN (Figure 1d and h).

The BenCS has larger physical biases in CESM-LENS than all other EBUS. Its SST bias is in excess of $7^oC$ with the nominal $1^o$ atmospheric resolution, compared to less than a $1^oC$ bias in the CalCS and CanCS, and a $1–3^oC$ bias in the HumCS (pers. comm. with RJ Small, 2018). This bias is likely driven by the fact that the Angola-Benguela Front is simulated too far south, in addition to deficiencies in upwelling and meridional transport that are caused by unrealistic alongshore wind stress structure (Small et al., 2015). Because these deficiencies are specific to the BenCS, we will only discuss its representation of the $pCO_2$ seasonal cycle in Section 3.1.2, and its internal variability in $CO_2$ fluxes in Section 3.2, but will not perform a full analysis on its connections to larger-scale climate variability.

### 3.1.2 $pCO_2$ Seasonal Cycle

CESM-LENS simulates the $pCO_2$ seasonal cycle well for the Pacific EBUS, with larger error in the Atlantic EBUS. In the CalCS, CESM-LENS nearly perfectly matches the SOM-FFN in both amplitude and phase (Figure 2a). The system exhibits its maximum $pCO_2$ (and thus $CO_2$ outgassing) in August, and its minimum $pCO_2$ (and thus $CO_2$ uptake) in April. We further decomposed the model seasonal cycle into its thermal component (driven by the seasonality of SST) and its non-thermal component (driven by the seasonality of factors such as DIC, ALK, and salinity) following Takahashi et al. (2002). This decomposition suggests that the phase of the CalCS seasonal cycle is determined by thermal (solubility) effects, with its amplitude modulated by non-thermal factors (Figure 2a). These non-thermal factors are almost entirely driven by the seasonal cycle of DIC (not shown), which is characterized by photosynthetic uptake of $CO_2$ in the summer and fall, coinciding with upwelling season. In the HumCS, both CESM-LENS and SOM-FFN suggest a dual peak in the seasonal cycle of $pCO_2$, although the model and observational product slightly disagree in phase and amplitude (Figure 2b). These two peaks are driven by an alternating dominance of thermal and non-thermal effects. During the austral summer/fall, warm temperatures lead to enhanced $pCO_2$ that is slightly compensated for by increased biological activity, similar to the singular peak of the CalCS. In the austral winter/spring, intense upwelling of DIC-enriched waters and reduced biological activity that is slightly compensated for by cooler waters leads to a secondary $pCO_2$ peak (Kämpf and Chapman, 2016). The SOM-FFN suggests that the CanCS behaves similarly to the CalCS, with a single $pCO_2$ peak in late summer/fall that is in phase with the seasonal cycle of SST (Figure 2c). However, CESM-LENS simulates a damped seasonal cycle that is approximately 180 degrees out of phase

for $pCO_2$ in the CanCS that results from a delicate balance between thermal and non-thermal effects of similar magnitudes. Despite the thermal and non-thermal effects being in the proper phase for a northern hemisphere system, the SST seasonal cycle is too weak and the DIC seasonal cycle too strong in the CanCS in CESM-LENS. Lastly, we find that CESM-LENS simulates the BenCS $pCO_2$ seasonal cycle nearly 180 degrees out of phase with the SOM-FFN representation (Figure 2d). Similar to the
CanCS, the thermal and non-thermal effects are in the proper phase. However, there is a large bias in the magnitude of the DIC seasonal cycle, which overwhelms the thermal seasonality and drives the $pCO_2$ seasonal cycle entirely out of phase.

## 3.2   Internal Variability in Upwelling Systems

We emphasize the magnitude of internal variability in EBUS air-sea $CO_2$ fluxes in Figure 3 by showing the ensemble mean standard deviation of air-sea $CO_2$ flux anomalies (ensemble mean subtracted) at each location across the global ocean. Save
for the Southern Ocean and subpolar Arctic, the EBUS emerge as regions of high internal variability on a global scale. The HumCS, CanCS, and BenCS in particular have some of the highest unforced variability in $CO_2$ fluxes globally. The CalCS has comparatively low internal variability in $CO_2$ fluxes. The EBUS generally have higher internal variability than other coastal regions and are particularly distinct from the major western boundary currents, which appear to be influenced very little by internal variability (Figure 3). The internal variability can also be isolated as one constituent of the area-weighted $CO_2$ flux time
series for each of the four EBUS, the others being the forced trend and seasonal cycle (Figure 4 (a–d)). The largest absolute internal variability component of the standard deviation of $CO_2$ flux is found in the BenCS ($0.98$ mol m$^{-2}$ yr$^{-1}$) and the HumCS ($1.20$ mol m$^{-2}$ yr$^{-1}$; Table 1). The BenCS is uniquely exposed to variability from the Southern Ocean and Agulhas Current (Reason et al., 2006). The HumCS likely has intense variability due to its proximity to the tropical Pacific Ocean and thus rapid communication with ENSO (e.g., Colas et al., 2008; Montes et al., 2011).

All four systems have statistically significant trends toward a weaker $CO_2$ source or greater $CO_2$ sink over 1920–2015 due mainly to the invasion of anthropogenic carbon from the atmosphere (Figure 4; Table 1). Note that in the CanCS and BenCS, there is a trend toward more outgassing of natural $CO_2$, which is compensated for by the relatively large uptake tendency of anthropogenic $CO_2$ (Table 1). The long-term trend forces the HumCS, CanCS, and BenCS to act as intermittent seasonal sinks by 2015 in some realizations due to the combination of the long-term trend and internal variability (Figure 4). The BenCS and
CanCS have the largest uptake of anthropogenic $CO_2$ over the historical period, although the HumCS has a relatively large uptake of both natural and anthropogenic $CO_2$ (Table 1). For the HumCS and BenCS, the contemporary $CO_2$ flux trend is on the order of the magnitude of their seasonal cycles over the course of 96 years. The CanCS is a unique case, where the contemporary trend is more than double the magnitude of its seasonal cycle (Table 1).

The magnitude of internal variability in the combined natural and anthropogenic $CO_2$ fluxes is greater than that of the
seasonal cycle for the majority of systems. The non-seasonal component of the total variability (the sum of the seasonal and internal components) is 59% for the HumCS, 73% for the CanCS, and 56% for the BenCS (Table 1). Only the CalCS has a stronger seasonal cycle of $CO_2$ flux than internal variability, but the non-seasonal component still accounts for 33% of the total variability in this system (Table 1). Lastly, internal variability in $CO_2$ fluxes tends to be phase-locked with the seasonal cycle, as the peak magnitudes of internal variability track the ridges and troughs of the seasonal component (Figure 4).

### 3.3 Climate Variability and Air-sea $CO_2$ Fluxes

Our primary goal for each EBUS was to identify the mode of climate variability most strongly associated with its $CO_2$ flux anomalies. We correlated area-weighted anomalies from the black boxes in Figure 1e–h for each simulation with every grid cell globally for a set of predictor variable anomalies: SST, sea level pressure (SLP), 10m wind speed, and wind stress curl. We then assessed the ensemble mean of the correlations to determine the mode of climate variability associated with the given global spatial pattern. Figure 5 displays one ensemble mean correlation case for the CalCS (a), HumCS (b), and CanCS (c) as well as violin plots showing the spread of correlations across the 34-member ensemble for Pacific (Figure 5d–e) and Atlantic (Figure 5f) modes of variability.

#### 3.3.1 California Current

The correlation between CalCS $CO_2$ flux anomalies and SSTa yields a map suggestive of Pacific Decadal Variability, due to the zonal dipole of correlations in the North Pacific (Figure 5a; Mantua and Hare, 2002; Di Lorenzo et al., 2008). Although similar in structure to the PDO, this map most closely resembles the NPGO (Di Lorenzo et al., 2008). In fact, correlations between the model-based NPGO with annual smoothing and CalCS $CO_2$ flux yields a correlation coefficient of -0.49 $\pm$ 0.04. In comparison, linear correlations with the PDO result in a correlation coefficient of 0.24 $\pm$ 0.05 (Figure 5d). Thus, we highlight the NPGO as the major mode of climate variability associated with anomalous $CO_2$ flux in the CalCS.

We find that the CalCS has a single-signed response to the NPGO with anomalous uptake of $CO_2$ during a positive event, intensifying the mean state of the system as an uptake site (Figure 6). The direct regression of $\Delta F$ onto the NPGO results in an anomalous uptake of 0.10 mol m$^{-2}$ yr$^{-1}$ $\sigma^{-1}$ (where "$\sigma$" refers to one standard deviation of the normalized principal component time series for the NPGO; Table 2), which is roughly 24% of the long-term historical $CO_2$ flux mean of -0.42 mol m$^{-2}$ yr$^{-1}$. The primary contributions to this uptake anomaly come from variations in SST and sDIC, which are mainly driven by changes to offshore gyre dynamics. As the oceanic expression of the North Pacific Oscillation, the NPGO is associated with intensified geostrophic circulation, resulting in increased transport in both the California and Alaskan Coastal Currents (Di Lorenzo et al., 2008). Further, a positive NPGO leads to enhanced upwelling-favorable winds south of Cape Mendocino (Figure S1). During the positive NPGO phase, the entire CalCS cools, increasing the $CO_2$ solubility (Figure 6, Figure S1). Although downwelling is enhanced offshore, increased DIC transport from the Alaskan Gyre leads to a tendency for outgassing of $CO_2$ (Figure S1). Nearshore south of Cape Mendocino, increased upwelling of DIC-enriched waters is compensated for by photosynthesis, leading to near-zero $CO_2$ flux anomalies (Figure S2). North of Cape Mendocino, weakened upwelling and low subsurface DIC anomalies lead to enhanced uptake anomalies (Figures S1, S2). Because the system-wide contributions of SST and sDIC to the anomalous flux nearly balance each other, minor contributions from wind, salinity, sAlk, and freshwater flux push the system in favor of anomalous uptake (Figure 6).

The CalCS has the largest relative ensemble spread in sDIC and sAlk (Figure 6; Table 2). This is potentially because of inter-simulation variability in the response of CalCS dynamics to the NPGO due to atmospheric noise (as in the case of ENSO in Deser et al., 2017, 2018) which can directly alter the biogeochemical properties of source waters that feed the region (Pozo Buil

and Di Lorenzo, 2017). Although the linear Taylor expansion approximates a $CO_2$ flux anomaly nearly half that of the direct regression of $\Delta F$ onto the NPGO, it is still of the same sign. This discrepancy is due to the influence of higher-order and cross-derivative terms that we did not account for in our linear approximation.

We also performed this analysis for the CalCS response to a $1\sigma$ positive (warm) phase of the PDO (Figure 7a and b). Every ensemble member displayed a dipole response to the PDO (not shown), with anomalous uptake in the nearshore region south of Cape Mendocino, and anomalous outgassing elsewhere in the domain (Figure 7c). This was the only case in which we found a non single-signed response across all ensemble members to any mode of climate variability investigated. However, note that the dipole pattern is quite similar to that of the CalCS response to the NPGO (Figure 6b). Both the nearshore and offshore regions have modest correlations with the PDO, with correlation coefficients of $-0.16 \pm 0.03$ and $0.28 \pm 0.05$, respectively. The positive phase of the PDO results in anomalously warm SSTs along the CalCS and causes weaker and shallower upwelling cells with higher retention of nutrient- and carbon-depleted surface waters (Figure S3; Chhak and Di Lorenzo, 2007). This aligns with the inverted contributions of SST and sDIC in Figure 7a and b relative to the contributions of these terms in response to the NPGO (Figure 6a). The warming of CalCS SSTs during a positive phase of PDO causes a reduction of $CO_2$ solubility and thus a tendency toward outgassing (Figure 7). Anomalous poleward coastal winds associated with the positive phase of the PDO result in reduced coastal and curl-driven upwelling along the entire coastline, and weakened curl-driven downwelling offshore (Figure S3). In coordination with reduced subsurface DIC throughout the region, these changes in circulation contribute toward anomalous uptake of $CO_2$ throughout the CalCS. Note that the nearshore decomposition in Figure 7a has a y-axis range four times smaller than that of the offshore decomposition. This slight uptake anomaly is the result of a delicate balance of minor terms, where the sDIC reduction slightly outweighs the warming effect. On the other hand, the offshore region has contributions from SST and sDIC that are as much as triple the magnitude as that for the NPGO (Table 2). Despite the sDIC reduction being larger than the SST term, the reduced sAlk is substantial enough to cause a slight outgassing anomaly offshore (Figure 7b).

The direct response of winds to the NPGO and PDO plays a negligible role in influencing anomalous $CO_2$ flux in the CalCS (Table 2). Although $\Delta U$ in response to the NPGO and PDO is on the order of the HumCS and CanCS, $\frac{\partial F}{\partial U}$ is 3–10 times smaller than the other systems. $\frac{\partial F}{\partial U}$ is based on the climatological mean U, $\Delta pCO_2$, and Schmidt number. The CalCS has the smallest mean $\Delta pCO_2$ of the EBUS – just $0.2\mu$atm. This causes $CO_2$ flux in the system to be relatively insensitive to fluctuations in the wind.

### 3.3.2 Humboldt Current

Global correlations between the HumCS $CO_2$ flux anomalies and SSTa display ENSO as a major influence, with regions of high correlation focused around the equatorial Pacific (Figure 5b). Correlations between HumCS $CO_2$ flux anomalies and the Nino3 index resulted in a correlation coefficient of $-0.40 \pm 0.04$ (Figure 5e). Similar results were found for the Nino3.4 index $(-0.38 \pm 0.04)$ and the Nino4 index $(-0.36 \pm 0.05)$. We chose the Nino3 index as our primary predictor of HumCS $CO_2$ flux anomalies, since it is more eastern-focused and thus captures the stronger spatial correlations closest to the HumCS (Figure 5b).

We present the results of a linear Taylor expansion for HumCS $CO_2$ flux anomalies regressed onto a $1^oC$ El Niño in Figure 8 (Equation 4). We find that the HumCS responds with a nearly single-signed $CO_2$ uptake anomaly, resulting in a weakening

of the climatological outgassing (Figure 8b). Although there is a small region in the northern HumCS that responds with an outgassing anomaly, it not nearly as coherent across the ensemble as was the spatial dipole response of the CalCS to the PDO (Figure 7c). The direct regression of $\Delta F$ onto the Nino3 index results in an anomalous uptake of 0.49 mol m$^{-2}$ yr$^{-1}$ K$^{-1}$, which is approximately 18% of the long-term historical $CO_2$ flux mean of 2.8 mol m$^{-2}$ yr$^{-1}$. As in the case of the CalCS, the two major terms contributing to the uptake anomaly are sDIC and SST, which are of opposite sign (Figure 8a). We would anticipate this to be the case, as an El Niño event induces warming along the HumCS as well as reduces the efficacy of upwelling for bringing carbon- and nutrient-rich waters to the surface due to the presence of an anomalously deep thermocline (Figure S4; Strub et al., 1998). While upwelling-favorable winds tend to decrease along Chile (outside of our study region) during an El Niño event, the coastal wind response along Peru is variable, ranging anywhere from slight downwelling-favorable to slight upwelling-favorable anomalies (Wyrtki, 1975; Enfield, 1981; Huyer et al., 1987).

In CESM-LENS, the HumCS experiences a warming of 0.7$^o$C for a 1$^o$C El Niño, which results in an outgassing tendency of 0.3 mol m$^{-2}$ yr$^{-1}$ K$^{-1}$ (Table 2). However, sDIC in the system is reduced by 13.2 mmol m$^{-3}$ K$^{-1}$ for the same event, which translates to a large uptake contribution of 0.8 mol m$^{-2}$ yr$^{-1}$ K$^{-1}$ (Table 2). This is an enormous change in sDIC, which is partially driven by the high subsurface DIC bias in the east equatorial Pacific in CESM1 (see Lovenduski et al., 2015, their Figure 2). The large sDIC reduction is due to weakened upwelling and a deepening of the thermocline by warm water advected from the equatorial Pacific (Figure S4). Although the passage of coastally trapped waves are important in the HumCS, they are not resolved in the coarse resolution CESM-LENS. Lastly, there is a minor outgassing anomaly of 0.06 mol m$^{-2}$ yr$^{-1}$ K$^{-1}$ in response to a slight increase in wind speed during El Niño (Table 2). Despite the significant contributions of wind speed, SST, and sAlk toward outgassing, the large reduction in sDIC drives an uptake anomaly that weakens the HumCS outgassing during an El Niño event. Note that these are the same mechanisms that are responsible for reduced outgassing in the tropical belt in response to an El Niño (Feely et al., 1999).

### 3.3.3 Canary Current

Correlations between the CanCS $CO_2$ flux anomalies and SLPa globally reveal a region of high positive correlation coefficients just northwest of Africa. This coincides with the climatological position of the Azores High, the large-scale anticyclone which forces the CanCS. The climate index that most directly captures variability in the Azores High is the NAO, and will thus be considered the main mode of climate variability that modulates anomalous $CO_2$ flux in the CanCS. We find modest correlations of $0.28 \pm 0.03$ between annually smoothed CanCS $CO_2$ flux anomalies and the NAO (Figure 5f). Relatively lower correlations are expected between anomalous EBUS $CO_2$ fluxes and atmospheric indices, as the atmosphere is noisier than the more slowly evolving ocean.

Grid cell correlations between CanCS $CO_2$ flux anomalies and the NAO are displayed in Figure 9b. The CanCS has a nearly single-signed response of increased outgassing during the positive phase of the NAO. The direct regression of $\Delta F$ onto a $1\sigma$ NAO results in an outgassing anomaly of 0.2 mol m$^{-2}$ yr$^{-1}$ $\sigma^{-1}$ (Table 2), which is 21% of the historical $CO_2$ flux mean of 0.95 mol m$^{-2}$ yr$^{-1}$. As with the other EBUS, the major contributors toward this anomaly are sDIC and SST (Figure 9a). The NAO represents fluctuations in the intensity of atmospheric circulation between the Azores High and Icelandic Low (Hurrell

et al., 2001). During the positive phase of the NAO, a stronger Azores High leads to intensified alongshore winds and thus more vigorous upwelling (Figure S5). This brings up additional deep cold water which in turn increases the $CO_2$ solubility of the system, tending toward an uptake anomaly of 0.15 mol m$^{-2}$ yr$^{-1}$ $\sigma^{-1}$ (Table 2). On the other hand, the increased sDIC from intensified upwelling is double the magnitude of the SST contribution, despite the increased biological activity which reduces

some of the physical sDIC input, leading to an outgassing anomaly of 0.33 mol m$^{-2}$ yr$^{-1}$ $\sigma^{-1}$. This large sDIC response is driven both by a high $\Delta sDIC$ of 3.9 mmol m$^{-3}$ $\sigma^{-1}$ as well as the fact that the CanCS has the highest $\frac{\partial F}{\partial sDIC}$ of the major EBUS. Increased winds of 0.3 m s$^{-1}$ $\sigma^{-1}$ lead to a significant outgassing pressure of 0.05 mol m$^{-2}$ yr$^{-1}$ $\sigma^{-1}$. This is due both to a large system sensitivity, $\frac{\partial F}{\partial U}$, to changes in wind and a high wind anomaly in response to the NAO. Ultimately, intensified winds and an anomalous increase in sDIC due to enhanced upwelling counteracts the solubility effects of colder SSTs and

increased photosynthetic uptake of sDIC.

## 4   Discussion and Conclusions

We find that the seasonal cycle of $p$CO$_2$ is single-peaked and its phase is driven by thermal (solubility) effects in the CalCS, CanCS, and BenCS, while non-thermal effects modulate its amplitude. The seasonal cycle of $p$CO$_2$ in the HumCS is dual-peaked and driven by an alternating dominance of thermal and non-thermal effects. Although CESM-LENS models the CalCS

and HumCS $p$CO$_2$ seasonal cycle well, the CanCS seasonal cycle is damped and both the CanCS and BenCS seasonal cycles are nearly 180 degrees out of phase (Figure 2).

Variations in sDIC and SST in response to large modes of climate variability exert the most influence on anomalous CO$_2$ fluxes in the CalCS, CanCS, and HumCS. Further, these terms always act in opposition to one another in the portions of variability associated with the climate indices explored in this study. Secondary to these terms are wind speed and sAlk.

Although their contributions do not rival those of SST and sDIC in magnitude, they act to further reinforce anomalies or to tip the balance toward outgassing or uptake when the SST and sDIC associated terms are of about equal magnitude. In all systems, salinity and freshwater fluxes have negligible contributions toward the total CO$_2$ flux anomaly.

CalCS and CanCS CO$_2$ flux anomalies are associated mainly with climate modes related to large-scale anticyclones, with their mean states (uptake for the CalCS and outgassing for the CanCS) intensified during phases of enhanced gyre circulation

in the NPGO and NAO, respectively. CO$_2$ flux anomalies in the HumCS are mostly driven by ENSO, due to its close proximity to the equatorial Pacific. We find that the HumCS has weakened outgassing during El Niño due to reduced upwelling and a deepening of the thermocline, which are similar mechanisms to the equatorial Pacific's response to El Niño (Feely et al., 1999).

Because sDIC anomalies are so critical to the EBUS response to internal climate variability, we further decomposed these anomalies into their absolute (Table 3) and relative (Figure 10) contributions from circulation (e.g., upwelling, advection, mix-

ing), biology (photosynthesis, respiration, and calcium carbonate formation and dissolution), and air-sea CO$_2$ fluxes. Changes in biological activity in the HumCS in response to ENSO contribute to nearly 50% (8.61 TgC yr$^{-1}$ K$^{-1}$) of the total sDIC tendency anomaly (Figure 10, Table 3). During an El Niño (La Niña) enhanced respiration (photosynthesis) pumps sDIC into (out of) the upper water column. While biology is the major contributor to sDIC tendency anomalies in the HumCS, circulation

changes play a leading role in the CalCS. In response to the NPGO, circulation anomalies in the CalCS contribute roughly 47% (1.47 TgC yr$^{-1}$ $\sigma^{-1}$) of the total sDIC tendency anomaly (Figure 10, Table 3). It is difficult to assess the relative contributions of $J'_{circ}$ and $J'_{bio}$ in the CanCS, as the large ensemble spread in $J'_{bio}$ drives a highly uncertain $J'_{circ}$, which is computed as the anomaly of the other three terms (Figure 10). In all three systems, CO$_2$ flux anomalies play an important role in modifying

the sDIC tendency, with an ensemble mean relative contribution greater than 25% (Figure 10). Note, however, that significant (moderate) spatial variability exists for circulation and biology contributions in the HumCS (CanCS), while contributions are relatively uniform throughout the CalCS (Figure S7).

It is important to note that anthropogenic climate change will likely modify our findings over the coming decades in a number of ways. The long-term addition of anthropogenic carbon to the surface ocean causes EBUS to become greater sinks for CO$_2$.

This trend shifts the mean state of the EBUS, causing historical outgassing sites (the HumCS, CanCS, and BenCS) to become intermittent net sinks of CO$_2$ under the influence of internal variability by the end of the historical period (Figure 4). Projecting to 2100 under RCP8.5 forcing, we find that the CanCS and BenCS become net sinks for CO$_2$ due to the anthropogenic trend, with mean CO$_2$ uptake of -0.43 mol m$^{-2}$ yr$^{-1}$ and -0.04 mol m$^{-2}$ yr$^{-1}$ over 2016–2100, respectively. The addition of anthropogenic CO$_2$ into the surface ocean can also intensify the magnitude of the seasonal cycle of CO$_2$ flux. This is due to the

increased concentration of CO$_2$ in the surface ocean as well as the reduction in the ocean's buffer capacity, which makes pCO$_2$ more sensitive to seasonal fluctuations in DIC and alkalinity (Landschützer et al., 2018, and references therein). This effect has been shown in observations (Landschützer et al., 2018) and is also projected in climate models (Kwiatkowski and Orr, 2018), but is only seen significantly in the CalCS and CanCS in CESM-LENS, with an approximate 37% and 30% increase in the CO$_2$ flux seasonal cycle over 2016–2100, respectively. Negligible changes to the seasonal cycle over the next century are projected

for the HumCS and BenCS in CESM-LENS.

External forcing will also alter the dynamics of the EBUS (Bakun et al., 2015; García-Reyes et al., 2015), potentially inducing changes to alongshore winds (Narayan et al., 2010; Sydeman et al., 2014; Oerder et al., 2015; Rykaczewski et al., 2015; Wang et al., 2015) and upper ocean stratification (Di Lorenzo et al., 2005; Rykaczewski and Dunne, 2010; Oerder et al., 2015), which will in turn influence the rate and efficacy of upwelling. It is unlikely that changes to vertical transport will

be detectable until mid-century, as the anthropogenic signal is obscured by internal variability (Brady et al., 2017). However, externally forced trends in stratification are likely to emerge much quicker (Henson et al., 2016). The biogeochemical signature of waters feeding upwelling (e.g., O$_2$, CO$_2$, and nutrient concentrations) will likely also be modified due to these dynamical changes as well as to changes to ocean ventilation and source water pathways (Rykaczewski and Dunne, 2010). Externally forced changes in physical and biogeochemical properties of EBUS will likely alter the relative contributions of different

physical processes to anomalous CO$_2$ fluxes, as approximated by Equation 4 and shown in Figures 6–9 and Table 2. It is also possible that modes of climate variability will change in response to anthropogenic forcing. Model projections and long-term observations have suggested that the intensity, frequency, or variance associated with ENSO (e.g., Timmermann et al., 1999; Cai et al., 2014, 2015), the NAO (Kuzmina et al., 2005), and the NPGO (Sydeman et al., 2013) has changed significantly in recent decades or will change over the next century. These modifications to modes of climate variability suggested by the

literature could directly impact the response of EBUS CO$_2$ flux anomalies to internal variability, thus affecting the conclusions

of our study. Further investigation of the impacts of anthropogenic climate change on $CO_2$ fluxes in EBUS is necessary for the community.

This study serves as a starting point toward better understanding how internal climate variability modulates $CO_2$ fluxes in the major EBUS. We present the mode of climate variability that has the largest influence on anomalous $CO_2$ fluxes in the HumCS and CanCS, and the leading two modes for the CalCS. Because of this, we account for approximately 16% of the total $CO_2$ flux variance in the HumCS and 8% in the CanCS. Since we analyzed statistically orthogonal modes (the PDO and NPGO) in the CalCS, we account for as much as 31% of the total $CO_2$ flux variance. It is difficult to explain a large chunk of the remaining variance for the EBUS, as other modes of internal variability are physically dependent on one another. Further, locally generated atmospheric noise in the EBUS can contribute substantially to the total modeled $CO_2$ flux variance. Previous studies that relate anomalous $CO_2$ fluxes to internal climate variability have explained similar amounts of variance. For example, Lovenduski et al. (2007) explains anywhere from 2% to 31% of $CO_2$ flux variance in the Southern Ocean in response to the Southern Annular Mode (SAM), McKinley et al. (2006) from 6% to 38% in the North Pacific in response to the PDO, and Gruber et al. (2002) 15% in the North Atlantic in response to the NAO.

Our study is limited by our use of a single coarse-resolution model ensemble, which carries uncertainty due to structural biases in the representation of the climate system and biogeochemistry, the ensemble's ability to accurately simulate the magnitude and frequency of internal climate variability, and processes that occur at a finer scale than the grid resolution. The community would benefit from future studies involving multiple single-model ensembles, which would reduce uncertainty due to structural biases, such as in the dynamics of the BenCS and the elevated sub-surface DIC concentration in the east equatorial Pacific in CESM-LENS. Due to model resolution, we do not resolve the coastal upwelling that induces vigorous outgassing within the first ∼50km of the coastline, such as in high-resolution model solutions by Turi et al. (2014) and Fiechter et al. (2014). This problem could be mitigated by nesting high-resolution EBUS simulations within a coarser global ensemble or by using regional mesh refinement techniques. This would allow the remote propagation of climate variability into the EBUS, while avoiding the high computational cost of running multiple high-resolution global simulations. In particular, the BenCS requires significant attention. We find pronounced internal variability in $CO_2$ fluxes in the BenCS in CESM-LENS that warrants investigation in a high-resolution model specific to the BenCS. We anticipate that these results and further investigation of the relationship between internal climate variability and anomalous $CO_2$ fluxes in EBUS will be useful for the rapidly developing subseasonal to decadal prediction community. Skillful prediction of climate variability, such as ENSO, the NPGO, and NAO, could be linked directly to anomalous fluxes of $CO_2$ in the major EBUS. As these systems are naturally sensitive to the undersaturation of calcium carbonate, these predictions could also aid in detecting and managing the onset of seasonal ocean acidification.

*Data availability.* The output from CESM-LENS is available as single variable time series at monthly, daily, and 6-hourly resolution through the Earth System Grid. Instructions for access and a full listing of available variables can be found under the UCAR Large Ensemble Project web page. Also on the CESM-LENS project page is the Climate Variability Diagnostics Package (CVDP), which includes the climate indices

used in this study for every simulation. Instructions for accessing the NPGO index for each simulation as well the code used to generate it can be found at http://www.cesm.ucar.edu/projects/community-projects/LENS/projects/npgo.html.

*Author contributions.* RXB and NSL designed the study. RXB analyzed the data, prepared figures and tables, and wrote the paper. NSL, MAA, MJ, and NG provided invaluable feedback throughout the study and reviewed the paper.

5  *Competing interests.* The authors of this study are unaware of any conflict of interest.

*Acknowledgements.* The National Science Foundation sponsors the National Center for Atmospheric Research where the Community Earth System Model is developed. Computing resources were provided by NCAR's Computational and Information Systems Laboratory. The Department of Energy's Computational Science Graduate Fellowship supported RXB throughout this study (DE-FG02-97ER25308). NSL and RXB are grateful for support from NSF (OCE-1752724, OCE-1558225). NG is grateful for support from the SNF project CALNEX
10  (grant no.149384) and ETH Zürich. We thank A. Phillips for his extensive work on developing the Climate Variability Diagnostics Package. We are grateful to two anonymous reviewers for their suggestions which helped in improving this manuscript. K. Karnauskas, J. Small, and M. Long provided feedback on earlier versions of this manuscript.

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

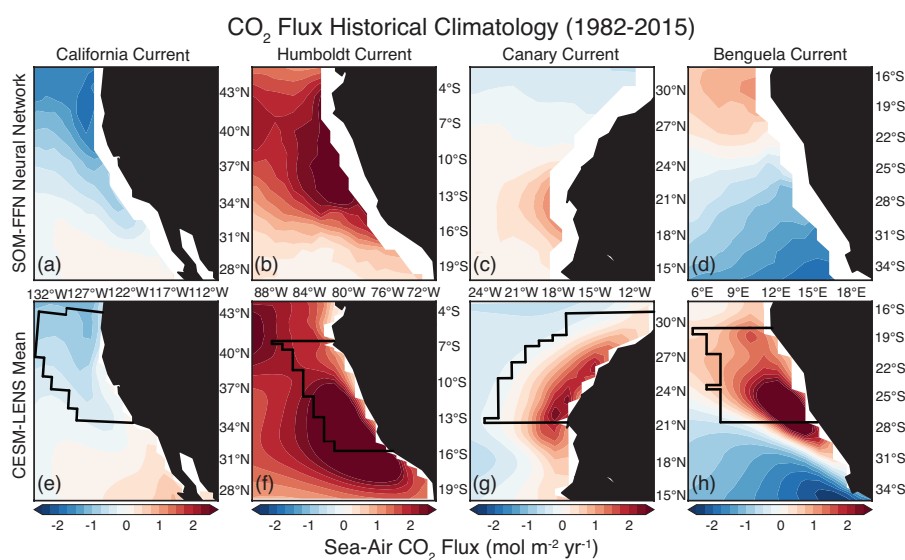

**Figure 1.** Comparison of $CO_2$ flux climatology from 1982–2015 between the SOM-FFN (a–d) and the CESM-LENS (e–h). Red denotes outgassing of $CO_2$ from the ocean to the atmosphere, while blue represents uptake of $CO_2$ by the ocean. Black lines in e–h follow the model grid and show the region used in each EBUS for statistical analysis, which is based on the $10^o$ latitude of most active upwelling from Chavez and Messié (2009) and confined to 800km offshore in the E–W direction.

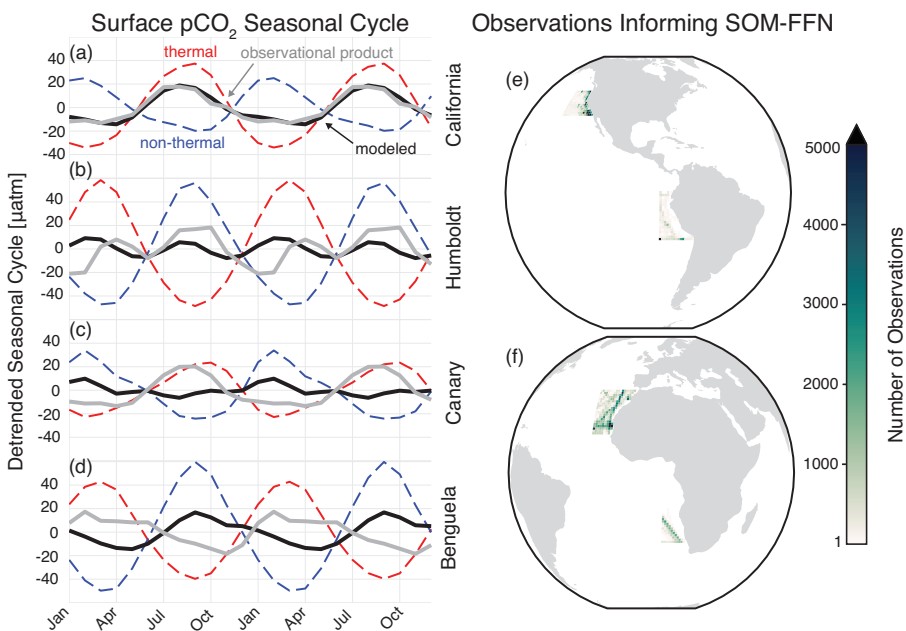

**Figure 2.** Detrended surface pCO$_2$ seasonal cycle from 1982–2015 for the SOM-FFN (gray) and the CESM-LENS ensemble mean (black) for the (a) CalCS, (b) HumCS, (c) CanCS, (d) and BenCS. Dashed red lines show the thermal component of the seasonal cycle for the CESM-LENS and dashed blue lines show the non-thermal component. The total number of observations from SOCATv4 contributing to the SOM-FFN for the EBUS is shown in (e) and (f).

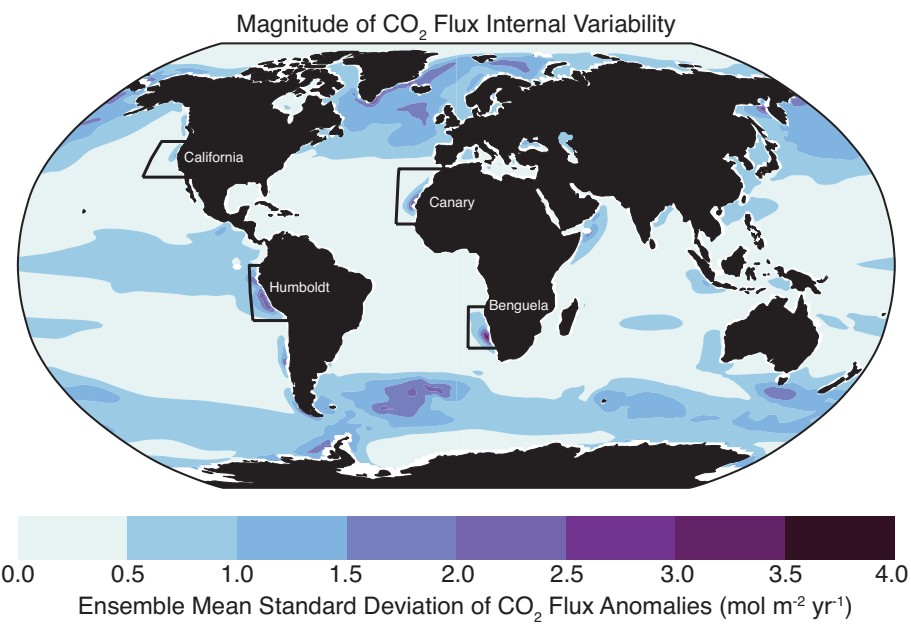

**Figure 3.** Magnitude of internal variability in $CO_2$ flux from 1920–2015 in the CESM-LENS. Anomalies were generated by removing the ensemble mean – which represents the seasonality and forced signal – from each realization. Internal variability was then quantified by taking the ensemble mean standard deviation of the anomalies from 1920–2015. Here, the black boxes outline the general domain of the EBUS in this study but do not coincide with the statistical boundaries shown in Figure 1.

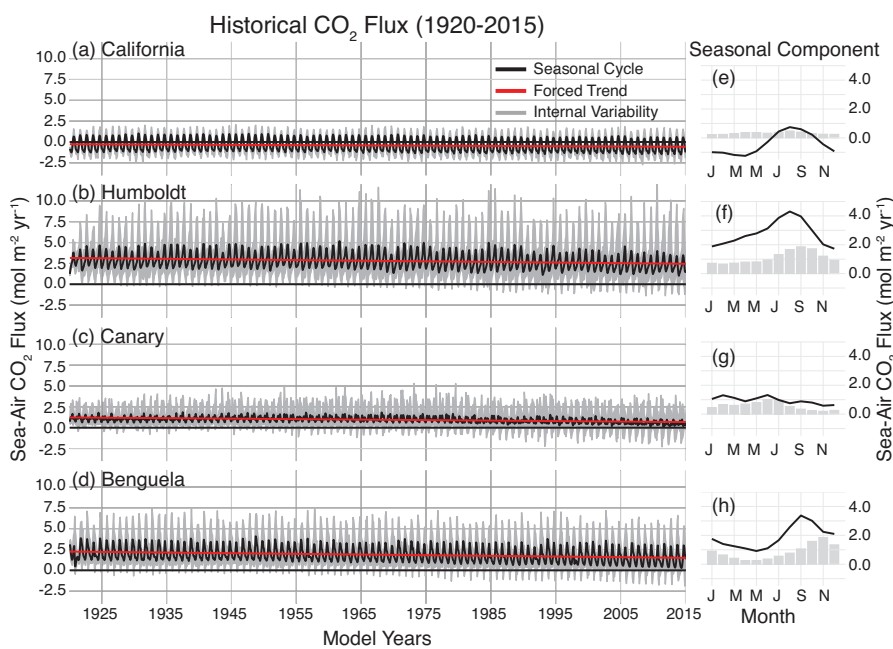

**Figure 4.** Time series of historical $CO_2$ flux (1920–2015) in the CESM-LENS for each of the four studied upwelling systems (a–d). The ensemble mean yields both the seasonal cycle (black) and the forced trend (red). Gray shading shows the bounds of the maximum and minimum realizations due to internal variability, but also contains signal from the seasonal cycle and forced trend. Table 1 displays the intercept, seasonality, internal variability, and forced trend for each system. Plots e–h show the mean seasonal cycle (black line) and ensemble mean monthly internal variability (gray bars) for 1920–2015 for each system.

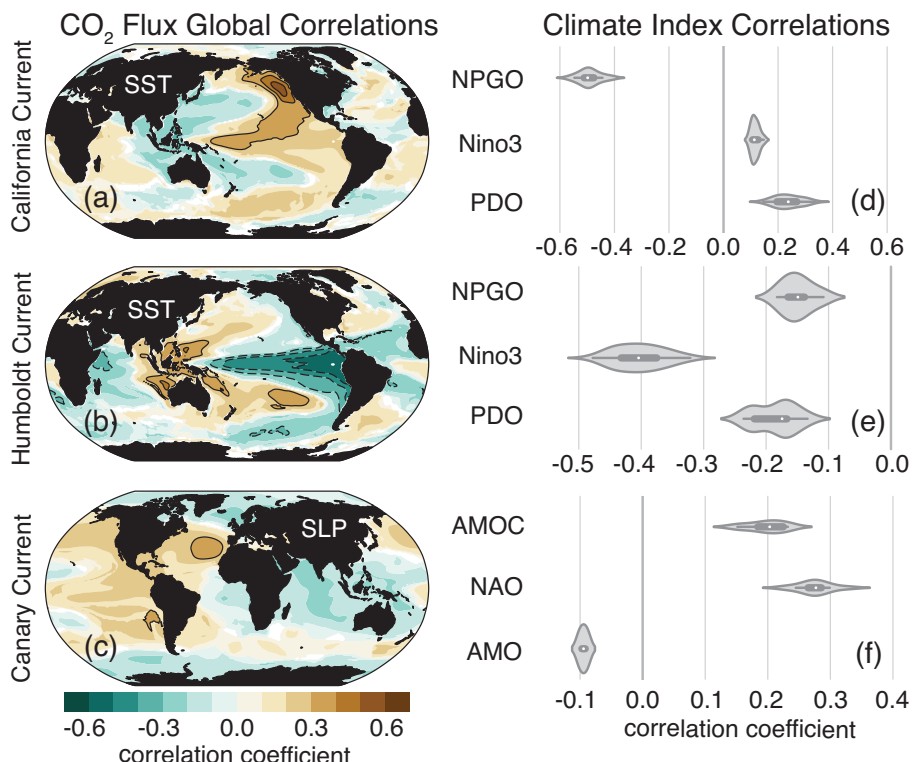

**Figure 5.** Correlations between area–weighted $CO_2$ flux anomalies in the statistical study regions outlined in black in Figure 1 (e–g) and SSTa (a–b; California, Humboldt) and SLPa (c; Canary) grid cells globally. Brown colors indicate that positive SSTa/SLPa correlates with outgassing, and blue with uptake. Contour lines begin at $\pm\ 0.3$ and progress in intervals of 0.1. Correlations were performed for each realization individually and the ensemble mean of those global correlations are presented here. Violin plots display correlations between area–weighted $CO_2$ flux anomalies from (a–c) with major modes of Pacific (d–e) and Atlantic (f) variability. The interior of the violin plot displays a box plot with the ensemble mean denoted as a white dot. Shading around the box plot reflects the ensemble distribution of correlations, which are mirrored on either side.

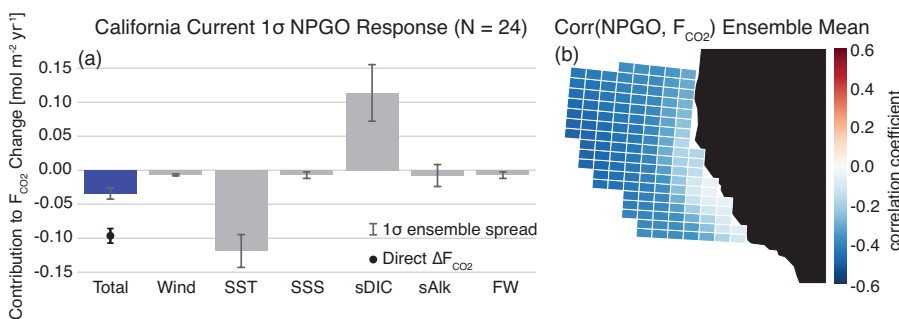

**Figure 6.** Linear Taylor expansion from Equation (4) for CalCS $CO_2$ flux anomalies regressed onto the NPGO (a). Gray bars represent the ensemble mean contributions of each variable to the $CO_2$ flux anomaly. Error bars represent the one standard deviation spread of the full ensemble. The individual bars sum to the "total" bar to approximate the direct regression of $\Delta F_{CO_2}$ onto the NPGO, which is denoted as the black dot with its associated ensemble spread. The ensemble mean grid cell correlations between $CO_2$ flux anomalies and the NPGO in the CalCS study region are displayed in (b). Positive correlations are associated with outgassing, negative with uptake. The magnitude and ensemble spread for each bar is presented in Table 2.

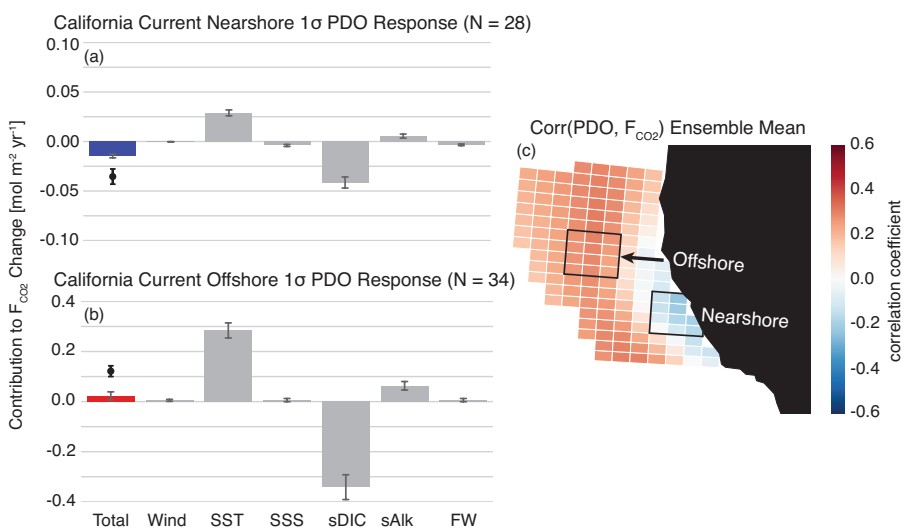

**Figure 7.** As in Figure 6, but in response to the PDO for a nearshore region (a) and offshore region (b). Note that the offshore decomposition (b) has a y-axis range four times that of the nearshore decomposition (a).

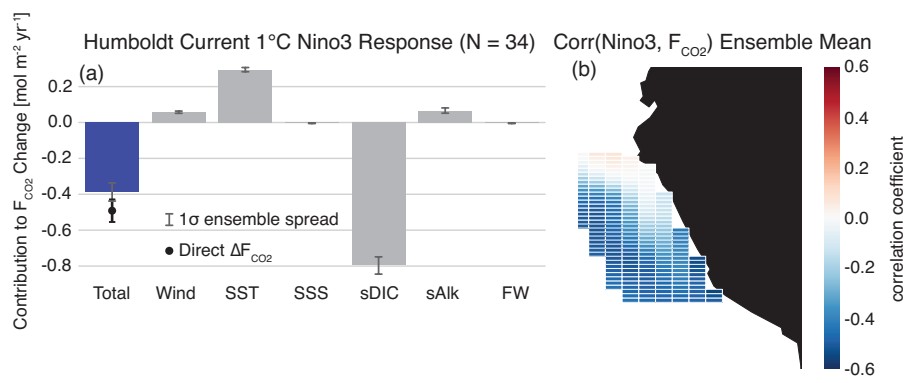

**Figure 8.** As in Figure 6, but for the Humboldt Current response to the Nino3 index.

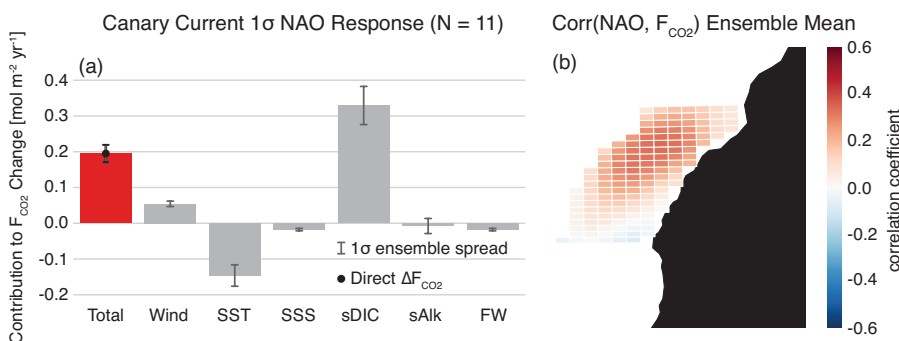

**Figure 9.** As in Figure 6, but for the Canary Current response to the NAO.

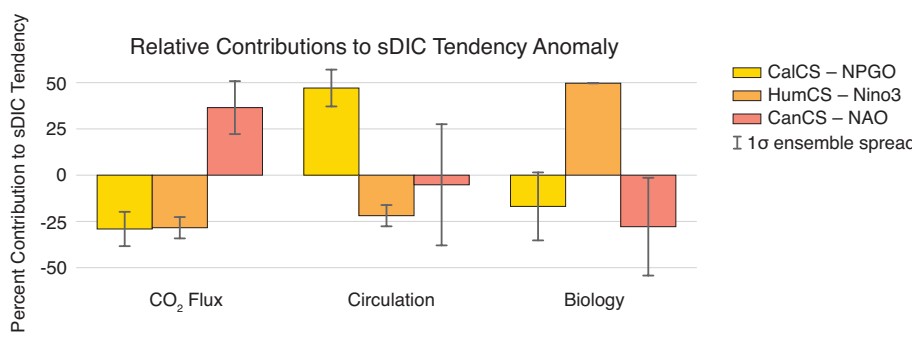

**Figure 10.** Relative contributions of anomalous air-sea $CO_2$ flux, physical circulation, and biology to the approximated sDIC tendency anomaly for the CalCS (yellow), HumCS (orange), and CanCS (red) in response to the NPGO, Nino3, and NAO, respectively. Error bars show the 1 standard deviation spread of the ensemble. For a given simulation, the absolute magnitude of the three components sum to 100%. These values were approximated using Equation 5.

**Table 1.** Statistics for air-sea $CO_2$ fluxes in the CalCS, HumCS, CanCS, and BenCS from 1920–2015. The seasonal component is computed as the standard deviation of the ensemble monthly climatology after removing a fourth-order polynomial fit to remove the long-term trend. The internal component is computed as the ensemble mean standard deviation of the anomalies. The trend is computed as the first-order ordinary least squares regression coefficient for the contemporary (total), anthropogenic, and natural $CO_2$ fluxes. The intercept is derived from this linear regression. The non-seasonal component represents the fraction of total variability (seasonal plus internal) to which the internal component contributes.

| Upwelling System | Intercept[1] | Seasonal[1] | Internal[1] | Trend (Contemporary)[2,3] | Trend (Anthropogenic)[2,3] | Trend (Natural)[2,3] | Non-Seasonal Component (%) |
|---|---|---|---|---|---|---|---|
| California ($34^o$ N–$44^o$ N) | -0.27 | 0.71 | 0.33 | -0.31 | -0.30 | -0.01 | 31 |
| Humboldt ($16^o$ S–$6^o$ S) | 3.16 | 0.83 | 1.20 | -0.71 | -0.67 | -0.04 | 59 |
| Canary ($21^o$ N–$31^o$ N) | 1.23 | 0.23 | 0.62 | -0.56 | -0.80 | 0.25 | 73 |
| Benguela ($28^o$ S–$18^o$ S) | 2.25 | 0.77 | 0.98 | -0.76 | -0.82 | 0.07 | 56 |

[1] $mol\ m^{-2}\ yr^{-1}$

[2] $mol\ m^{-2}\ yr^{-1}\ 96yr^{-1}$

[3] All trends are significant to $\alpha = 0.05$ for a one-sided Mann-Kendall Test

**Table 2.** Estimated contributions of individual terms to air-sea $CO_2$ flux anomalies, $\Delta F$, in response to a mode of climate variability using Equation 4. Each row under the *Individual Terms* header depicts the ensemble mean contribution and ensemble spread for Figures 6–9. The column "CalCS–PDOn" reflects results from the nearshore box in the CalCS in Figure 7, and "CalCS–PDOo" the offshore box. $\Sigma$ is the sum of all contributing first-order terms (i.e., all rows under *Individual Terms* or the right hand side of Equation 4). $\Delta F$ is the direct regression of $CO_2$ flux anomalies onto the specified mode of climate variability (i.e., the left hand side of Equation 4).

| Quantity | CalCS – NPGO[1] | CalCS – PDOo[1] | CalCS – PDOn[1] | HumCS – Nino3[2] | CanCS – NAO[1] |
|---|---|---|---|---|---|
| *Individual Terms* | | | | | |
| $\frac{\partial F}{\partial U}\Delta U$ | -0.01 ± 0.0 | 0.01 ± 0.0 | 0.0 ± 0.0 | 0.06 ± 0.01 | 0.05 ± 0.01 |
| $\frac{\partial F}{\partial T}\Delta T$ | -0.12 ± 0.02 | 0.28 ± 0.03 | 0.03 ± 0.0 | 0.29 ± 0.01 | -0.15 ± 0.03 |
| $\frac{\partial F}{\partial S}\Delta S$ | -0.01 ± 0.0 | 0.01 ± 0.01 | 0.0 ± 0.0 | -0.0 ± 0.0 | -0.02 ± 0.0 |
| $\frac{S}{S_0}\frac{\partial F}{\partial DIC}\Delta sDIC$ | 0.11 ± 0.04 | -0.34 ± 0.05 | -0.04 ± 0.01 | -0.8 ± 0.05 | 0.33 ± 0.05 |
| $\frac{S}{S_0}\frac{\partial F}{\partial Alk}\Delta sAlk$ | -0.01 ± 0.02 | 0.06 ± 0.02 | 0.01 ± 0.0 | 0.07 ± 0.01 | -0.01 ± 0.02 |
| $\frac{\partial F}{\partial fw}\Delta fw$ | -0.01 ± 0.0 | 0.01 ± 0.01 | 0.0 ± 0.0 | 0.0 ± 0.0 | -0.02 ± 0.0 |
| *Sum of Terms Versus Modeled* | | | | | |
| $\Sigma$ | -0.03 ± 0.01 | 0.02 ± 0.02 | -0.01 ± 0.0 | -0.38 ± 0.05 | 0.21 ± 0.03 |
| $\Delta F$ | -0.10 ± 0.01 | 0.12 ± 0.02 | -0.04 ± 0.01 | -0.49 ± 0.06 | 0.2 ± 0.02 |

[1] $\mathrm{mol\ m^{-2}\ yr^{-1}\ \sigma^{-1}}$

[2] $\mathrm{mol\ m^{-2}\ yr^{-1}\ K^{-1}}$

**Table 3.** Regression coefficients between the given EBUS and climate index for anomaly time series of the estimated contributions toward the sDIC tendency integrated over the upper 100m and the total surface area of the system, as in Equation 5.

| Term | CalCS – NPGO[1] | HumCS – Nino3[2] | CanCS – NAO[1] |
|------|-----------------|------------------|----------------|
| $\frac{ds DIC'}{dt}$ | $0.03 \pm 0.01$ | $-0.10 \pm 0.02$ | $0.04 \pm 0.01$ |
| $J'_{ex}$ | $-0.76 \pm 0.20$ | $-4.84 \pm 0.61$ | $0.51 \pm 0.21$ |
| $J'_{bio}$ | $-0.68 \pm 0.48$ | $8.61 \pm 0.81$ | $-0.49 \pm 0.54$ |
| $J'_{circ}$ | $1.47 \pm 0.64$ | $-3.87 \pm 1.28$ | $0.02 \pm 0.55$ |

[1] $\text{TgC yr}^{-1} \sigma^{-1}$
[2] $\text{TgC yr}^{-1} \text{K}^{-1}$