# Peer review of "On the role of climate modes in modulating the air-sea CO2 fluxes in Eastern Boundary Upwelling Systems"

_Biogeosciences, 2018_

## Referee Comment (RC1) · Anonymous Referee #1 · 22 Nov 2018

Review of manuscript bg-2018-415 "On the role of climate modes in modulating the air-sea CO2 fluxes in Eastern Boundary Upwelling Systems" by Riley X. Brady, Nicole S. Lovenduski , Michael A. Alexander , Michael Jacox , and Nicolas Gruber.

This is a nice piece of work trying to link modes of natural climate variability to fluctuations of air-sea CO2 fluxes in Eastern Boundary Upwelling Systems (EBUS). The latter regions reveal strong upwelling of cold and nutrient-reach water masses, a prerequisite for ecological richness and diversity. Although small in area, EBUSs play an important role in the air-sea exchange of CO2. Utilizing 34 ensemble members of simulations with the Community Earth System Model (CESM-LENS) the authors have analyzed

the correlations between natural climate variability (ENSO, PDO, NAO, . . .) and air-sea CO2 fluxes for each of the four EBUSs.

The manuscript is clearly written and the conclusions comprehensible. Therefore, subject to very minor revisions I recommend publication in Biogeosciences.

Specific comments:

(1) page 5 equation 2: The variable U is not defined (presumably wind speed).

(2) page 5 last sentence: "To compensate for autocorrelation ..." This should be explained a bit more in detail. Readers not familiar with the statistical methodology will not understand it.

(3) page 6 lines 18+19: " . . . the data density of pCO2 in EBUS informing the SOM-FFN is on the order of the Southern Ocean, ..." This statement is too vague. It would be good to be a bit more quantitative.

---

## Referee Comment (RC2) · Anonymous Referee #2 · 23 Nov 2018

I think this is a good paper that is publishable with minor to moderate revision. Some details of the methodology are insufficiently explained. The English is generally good, although there are some quirks of usage that suggest an inexperienced lead author whose more senior coauthors put rather less time into editing the text than they could have.

Overall structure:

I think that "Conclusions and Discussion" would be better entitled "Discussion and Conclusions", and Section 3 should be incorporated into the Results. The first paragraph of the Discussion covers a lot of different topics, and rehashes a lot of the Results. It

would be better to lead off with a summation of the main points, and then further discussion of each, broken into a larger number of shorter paragraphs each focused on a specific topic. The Introduction meanders about a number of related topics in a way that could easily give the reader the impression that high-resolution regional hindcast simulations were employed (3/15-20). There is nothing wrong with mentioning the utility of such tools, but ideally one should try to structure the Introduction in a way that focuses on (1) what is the problem at hand? (2) what tools were used to address it? and (3) what is novel in the analysis that sets it apart from what is already in the literature? Similarly, I don't think that the idea that previous EBC studies have not focused on CO2 (2/18, 3/8) is either accurate or relevant (and the sentence on 2/18-20 is simply ugly). Statements like 7/27-29 are also unnecessary; this section is properly part of the Results, and statements like this belong in the Discussion (see also 6/11-13, 7/10-11).

Methods

I don't think the definitions of the boundaries of the boxes are sufficiently explained. The outer boundaries of the boxes in Figure 3 are not parallel to the coastline, so "from the coastline to 800 km offshore" hardly seems adequate. Turi et al only use the 800 km figure in general terms, in reference to the approximate domain of influence of coastal upwelling on the thermocline depth. In Figure 1, the boxes seem to indicate an approximately (but not exactly) constant distance in the E-W direction. The boxes in Figure 3 are quite different. The regions considered in Figures 6-9 are similar, but not necessarily identical, to those in Figure 1. A clearer explanation is warranted, especially given that many of the analyses shown are for regional means over these boxes. (Note also that the second half of Section 2.2 has nothing to do with the topic specified in header.)

For the regressions onto the climate indices, it should be be more clearly stated what the independent variable and its units are. References to a "1 degree El Nino" or 1 SD of NAO are better than nothing but not really adequate. If NINO3 is defined as a temperature anomaly in K, then a regression coefficient of CO2 flux on this index will

have units of mol mˆ-2 yˆ-1 Kˆ-1 (e.g., 11/2). Similarly, the NAO has units of SLP. I'm not sure what the units of the NPGO are. But a statement like "The direct regression of DeltaF onto the NPGO results in an anomalous uptake of 0.10 mol mˆ-2 yrˆ-1" seems incomplete, because the reader does not know how large an anomaly in the NPGO is required to give rise to X mol mˆ-2 yrˆ-1 of CO2 flux anomaly. The statement (12/10) that stronger winds in the CanCS "leads to the highest relative CO2 flux anomaly of any system" seems misleading because the independent variable for these regressions is different in each case. Maybe there is some basis for making this comparison, but it has not been clearly explained.

The statistical tests applied are inadequately explained. The discussion of autocorrelation (5/27) appears out of nowhere without context. I agree that autocorrelation is important and you have to correct for it, but up to this point there has been no mention of statistical testing at all. First explain what test you are using to determine whether X is significantly different from 0, and state clearly what physical quantity X represents, then explain the effective sample size. The effective sample size is said to "replace the t-statistic sample size" (5/27), but there is no mention of t-tests having been conducted; the only test specifically mentioned is the Mann-Kendall test (Table 1).

Model evaluation

I find parts of the discussion of model validation against SOM-FFN confusing. On the one hand, it is the best observational benchmark available, on the other hand, discrepancies are explained away as resulting from errors in the observational data product (6/17-21). I can't make sense of "CO2 fluxes in SOM-FFN are being informed by remote biogeochemical provinces more often than other regions of the ocean". I can guess at what is being stated here, but without a more specific explanation of what sort of bias it imparts to the data product, I don't think it helps to achieve the task at hand, which is to evaluate how physically realistic the model solution is. Describing a model as 'biased' without specifying the nature or sign of the bias (e.g., 6/15, 6/21) is not very useful. The beginning of 3.2 is misleading (CalCS shows very good results, HumCS

much less so) and poorly worded. How about "CESM-LENS simulates the pCO2 seasonal cycle well for the Pacific EBUS, with larger error in the Atlantic regions"? (and change "Beginning with the CalCS" to simply "in the CalCS"). Again, this paragraph mixes up model evaluation, analyses of the model solution that do not have any analogue in the observations, and literature review. Again, I think this whole section should be included in the Results, and a clear separation of Results and Discussion attempted.

Terminology

"internal" variability in a model simulation is an analogue of "natural" climate variability in the real world. I would prefer that the latter term be used except when the reference is specifically to climate model simulations. I find terms like "have some of the highest internally driven CO2 fluxes globally" very awkward. How about "have some of the highest unforced variability of CO2 flux of any part of the world ocean"?

I think "values" is one of the most overused and abused words in scientific writing. Search out all occurrences and if possible replace with something specific. For example, on 10/8 one could replace "r-values" with "correlation coefficients" (see also 9/13-14, 11/22) and on 13/19 "mean values" could be "mean uptake". On 8/11, one can't even tell what physical quantity is being presented (it is the internal variability component of the standard deviation of the CO2 flux, but the reader has to go to the table caption to find this out).

There are many locations where "air-sea" could be added before "CO2 flux" in the interest of clarity (e.g., Figure 10 caption).

Some details

1/21 "Upwelling delivers deep waters with respired nutrients to the surface, fueling primary production and ultimately supporting fisheries that are highly productive with respect to the small surface area they cover" Upwelling delivers waters rich in nutrients to the surface, fueling primary production and ultimately supporting fisheries that are

highly productive relative to the small surface area that they cover

2/5 "contributing to the magnitude and determining the direction of air-sea CO2 fluxes" determining the sign and magnitude of the air-sea CO2 flux

2/11 "more efficient biology" greater biological production

2/21 delete "fractional"

3/5 Is it accurate to refer to the NAO as a "decadal-scale oscillation"? I thought it was more like a white noise process with a very flat frequency spectrum.

4/20-21 "In contrast ... CO2 fluxes." I would delete this entire sentence.

7/13 "These dual peaks are driven by an interchanging importance between thermal and non-thermal effects." These two peaks are driven by an alternating dominance of thermal and non-thermal effects. (see also 12/15 and 12/18)

7/22 "the BenCS pCO2 seasonal cycle nearly 180 degrees out of phase" This actually true of CanCS as well, although the amplitude is significantly underestimated in the model.

9/23 change "During a positive NPGO event" to "During the positive NPGO phase"? (and delete "of the system")

9/24 add "transport" after "DIC"

9/27 "Because the system-wide contributions of SST and sDIC to the anomalous flux nearly balance each other, minor contributions from wind, salinity, sAlk, and freshwater flux push the system in favor of anomalous uptake" I think this is an overinterpretation. It looks to me like the SST contribution is larger than the sDIC, and even if the 4 smaller terms cancelled each other out the net would still be negative.

10/27 change "influencer" to "influence"

11/4 change "are in opposition to one another" to "are of opposite sign"

11/14 change "advected warm waters from the equatorial Pacific" to "warm water advected from the equatorial Pacific"

11/17 change "intensification of wind magnitude" to "increase in wind speed"

11/23 "This encircles the climatological position of the Azores High, the atmospheric subtropical gyre which forces the CanCS." I have never heard the Azores High referred to as an "atmospheric subtropical gyre", although it is a large-scale anticyclone. But I have never heard this terminology before, and it's generally bad practice to take existing terms and assign them new meanings without a compelling reason. I also don't think "encircles" is a good choice of words. How about "represents" (or "indicates" or "coincides with")?

11/32 "the linear Taylor approximation aligns exactly with the direct regression" I can't tell what this means.

11/33 "The NAO describes modifications to the intensity of atmospheric gyre circulation between the Azores High and Icelandic Low" The NAO represents fluctuations in the intensity of atmospheric circulation between the Azores High and Icelandic Low

12/21 change "when sDIC and SST are of equal magnitude" to "when the sDIC and SST associated terms are of about equal magnitude"

12/27-30 "The major EBUS ... variability in $CO_2$ fluxes." another truly awful sentence: rewrite or delete

13/13 change "diffusion" to "mixing"

13/7 delete "to" before "roughly"

13/25 I think this statement requires a data or literature reference.

13/34 "the relative contributions of variables to anomalous $CO_2$ fluxes" the relative contributions of different physical processes to anomalous $CO_2$ fluxes

14/4 "While not observed in our historical modeling study, modifications to modes of climate variability associated with the major EBUS could directly influence the magnitude of internally generated anomalies in CO2 fluxes in the future." I don't see how we know this. Such trends might exist in the ensemble data even if no one has yet attempted to detect them.

14/9 "we only present the leading mode of climate variability" Similarly, this may be true but I don't think it is demonstrated by the data shown in this paper. The authors simply focus, in each region, on what they EXPECT to be the most important mode; they don't actually test whether this is true.

14/10-12 "we explain", "we were able to explain" not an appropriate use of first person (I suggest that the wording of all discussions of explained variance in this paragraph be reviewed.)

14/19 change "a coarse single model ensemble" to "a single coarse-resolution model"

14/24 "do not directly resolve the coastal upwelling process which induces vigorous outgassing within the first O(10km) of the coastline" do not resolve the coastal upwelling that induces vigorous outgassing within the first $\sim$10 km of the coast

14/28-29 I agree with this sentiment, but you have to get the boundary conditions for the downscaling model from global models. So if those models have huge biases in the positions of major transition zones, it's not clear that having high resolution within a regional domain is going to do any good. This is a problem in the northwest Atlantic as well, as coarse resolution models have large and persistent biases in the location of the Gulf Stream separation from the coast.

Figure 5 contains a great deal of information, and the caption could be a bit clearer. Violin plots may not be familiar to some readers, and exactly what is shown in the right hand panels could be spelled out. Similarly, the caption to Table 2 could contain a great deal more detail.

[Figure]

---

## Author Comment (AC1) · 14 Dec 2018

**Reviewer #1:**

*Review of manuscript bg-2018-415 "On the role of climate modes in modulating the air-sea CO2 fluxes in Eastern Boundary Upwelling Systems" by Riley X. Brady, Nicole S. Lovenduski, Michael A. Alexander, Michael Jacox, and Nicolas Gruber.*

**Summary:**

This is a nice piece of work trying to link modes of natural climate variability to fluctuations of air-sea CO2 fluxes in Eastern Boundary Upwelling Systems (EBUS). The latter regions reveal strong upwelling of cold and nutrient-reach water masses, a prerequisite for ecological richness and diversity. Although small in area, EBUSs play an important role in the air-sea exchange of CO2. Utilizing 34 ensemble members of simulations with the Community Earth System Model (CESM-LENS) the authors have analyzed the correlations between natural climate variability (ENSO, PDO, NAO, ...) and air-sea CO2 fluxes for each of the four EBUSs.

The manuscript is clearly written and the conclusions comprehensible. Therefore, subject to very minor revisions I recommend publication in Biogeosciences.

*We would like to thank referee #1 for their time in reviewing this paper. Their suggestions substantially improved this manuscript. Please see the supplemental pdf to this response for a tracked changes version of the revised manuscript.*

**Specific Comments:**

1. page 5 equation 2: The variable U is not defined (presumably wind speed).

   *Thank you for catching this error. We now define U as wind speed in page 5 line 10, since it is introduced as being a factor in calculating* k.

2. page 5 last sentence: "To compensate for autocorrelation ..." This should be explained a bit more in detail. Readers not familiar with the statistical methodology will not understand it.

   *Thank you for this suggestion. We agree that in its original form, the explanation*

*was not entirely clear. We have now updated the manuscript to the following:*

> *Autocorrelation is prevalent in climate indices such as the NPGO and ENSO (Di Lorenzo and Ohman, 2013), and our annual smoothing further enhances autocorrelation in CalCS and CanCS air-sea $CO_2$ fluxes (see Sections 3.3.1 and 3.3.3). To compensate for this autocorrelation, we replace the t-statistic sample size N with an effective sample size $N_{eff}$, which quantifies the number of statistically independent measurements: . . .*

3. page 6 lines 18+19: "... the data density of pCO2 in EBUS informing the SOM-FFN is on the order of the Southern Ocean, ..." This statement is too vague. It would be good to be a bit more quantitative.

*Thank you for this suggestion. We have quantified this by taking the mean number of observations informing the SOM-FFN for each EBUS (following the regions shown in Figure 2e and 2f) and for the Southern Ocean (south of 40S). Page 6 lines 18+19 were updated to the following:*

*Another important caveat is that the average number of $pCO_2$ observations in EBUS informing the SOM-FFN (637, 119, 517, and 195 for the CalCS, HumCS, CanCS, and BenCS, respectively) is on the order of the Southern Ocean (536), a notably undersampled region (Figure 2e and f; Bakker et al., 2016).*

**References**

Bakker, D. C. E., Pfeil, B., OBrien, K. M., Currie, K. I., Jones, S. D., Landa, C. S., Lauvset, S. K., Metzl, N., Munro, D. R., Nakaoka, S.-I., Olsen, A., Pierrot, D., Saito, S., Smith, K., Sweeney,

C., Takahashi, T., Wada, C., Wanninkhof, R., Alin, S. R., Becker, M., Bellerby, R. G. J., Borges, A. V., Boutin, J., Bozec, Y., Burger, E., Cai, W.-J., Castle, R. D., Cosca, C. E., De-Grandpre, M. D., Donnelly, M., Eischeid, G., Feely, R. A., Gkritzalis, T., González-Dávila, M., Goyet, C., Guillot, A., Hardman-Mountford, N. J., Hauck, J., Hoppema, M., Humphreys, M. P., Hunt, C. W., Ibánhez, J. S. P., Ichikawa, T., Ishii, M., Juranek, L. W., Kitidis, V., Körtzinger, A., Koffi, U. K., Kozyr, A., Kuwata, A., Lefèvre, N., Lo Monaco, C., Manke, A., Marrec, P., Mathis, J. T., Millero, F. J., Monacci, N., Monteiro, P. M. S., Murata, A., Newberger, T., No-jiri, Y., Nonaka, I., Omar, A. M., Ono, T., Padín, X. A., Rehder, G., Rutgersson, A., Sabine, C. L., Salisbury, J., Santana-Casiano, J. M., Sasano, D., Schuster, U., Sieger, R., Skjelvan, I., Steinhoff, T., Sullivan, K., Sutherland, S. C., Sutton, A., Tadokoro, K., Telszewski, M., Thomas, H., Tilbrook, B., van Heuven, S., Vandemark, D., Wallace, D. W., and Woosley, R. (2016). Surface Ocean CO2 Atlas (SOCAT) V4.

Di Lorenzo, E. and Ohman, M. D. (2013). A double-integration hypothesis to explain ocean ecosystem response to climate forcing. *Proceedings of the National Academy of Sciences*, 110(7):2496–2499.

---

## Author Comment (AC2) · 14 Dec 2018

**Reviewer #2:**

**Summary:**
I think this is a good paper that is publishable with minor to moderate revision. Some details of the methodology are insufficiently explained. The English is generally good, although there are some quirks of usage that suggest an inexperienced lead author whose more senior coauthors put rather less time into editing the text than they could

have.

*We would like to thank referee #2 for their careful review of this paper. Their suggestions substantially improved this manuscript. **Please see the supplemental pdf to this response for a tracked changes version of the revised manuscript.***

**Overall structure:**
I think that "Conclusions and Discussion" would be better entitled "Discussion and Conclusions", and Section 3 should be incorporated into the Results. The first paragraph of the Discussion covers a lot of different topics, and rehashes a lot of the Results. It would be better to lead off with a summation of the main points, and then further discussion of each, broken into a larger number of shorter paragraphs each focused on a specific topic.

*Thank you for your suggestion. We have changed the final section to be called "Discussion and Conclusions." We have also incorporated Section 3 into the Results.*

*We have split the first paragraph of Section 5 into three smaller paragraphs covering separate topics (i.e., the seasonal cycle discussion, the general $CO_2$ flux response to modes of climate variability, and the more specific $CO_2$ flux response to climate variability.)*

The Introduction meanders about a number of related topics in a way that could easily give the reader the impression that high-resolution regional hindcast simulations were employed (3/15-20). There is nothing wrong with mentioning the utility of such tools, but ideally one should try to structure the Introduction in a way that focuses on (1) what is the problem at hand? (2) what tools were used to address it? and (3) what is

novel in the analysis that sets it apart from what is already in the literature?

*Thank you for your suggestions. We have removed 3/15–20 and replaced it with a much briefer description of high-resolution simulations:*

> *Regional hindcast simulations are beneficial for their higher spatial resolution and more accurate representation of a specific EBUS's dynamics, but they are limited to the analysis of a single EBUS, preventing a synchronous view across EBUS with a consistent modeling tool.*

Similarly, I don't think that the idea that previous EBC studies have not focused on CO2 (2/18, 3/8) is either accurate or relevant (and the sentence on 2/18-20 is simply ugly).

*Thank you for your comments. We have edited 2/18–20 to read more cleanly:*

> *So far, relatively few studies have truly assessed the longer-term variability of the air-sea $CO_2$ fluxes in EBUS, regardless of whether these variations are internal or forced.*

*Indeed, very few EBC studies have focused directly on the link between climate variability and air-sea $CO_2$ fluxes (Chavez et al., 1999; Friederich et al., 2002; Torres et al., 2003; Feely et al., 2006; Takahashi et al., 2003), in comparison to the response of physics and biology to climate variability (e.g., Chenillat et al., 2012; Chhak and*

[Figure]

*Di Lorenzo, 2007; Di Lorenzo et al., 2008, 2009; Mantua et al., 1997; Cropper et al., 2014; Shannon et al., 1986; Reason et al., 2006; Hutchings et al., 2009; Chelton et al., 1982; Barber and Chavez, 1983; Barber and Chávez, 1986; Lynn and Bograd, 2002; Escribano et al., 2004; Frischknecht et al., 2015, 2017; Borges et al., 2003). We feel that mentioning this in 2/18 and 3/8 is useful to motivate the need for a study investigating air-sea $CO_2$ fluxes directly.*

Statements like 7/27-29 are also unnecessary; this section is properly part of the Results, and statements like this belong in the Discussion (see also 6/11-13, 7/10-11).

*Thank you for your suggestions. We excised 7/10–11 and 7/27–29 from the manuscript. We removed 6/11–13 and adapted 14/24–25 to read:*

> *Due to model resolution, we do not resolve the coastal upwelling that induces vigorous outgassing within the first ∼50km of the coastline, such as in high resolution model solutions by Turi et al. (2014) and Fiechter et al. (2014).*

**Methods:**
I don't think the definitions of the boundaries of the boxes are sufficiently explained. The outer boundaries of the boxes in Figure 3 are not parallel to the coastline, so "from the coastline to 800 km offshore" hardly seems adequate. Turi et al only use the 800 km figure in general terms, in reference to the approximate domain of influence of coastal upwelling on the thermocline depth. In Figure 1, the boxes seem to indicate an approximately (but not exactly) constant distance in the E-W direction. The boxes in Figure 3 are quite different. The regions considered in Figures 6-9 are similar, but not necessarily identical, to those in Figure 1. A clearer explanation is warranted,

especially given that many of the analyses shown are for regional means over these boxes. (Note also that the second half of Section 2.2 has nothing to do with the topic specified in header.)

*Thank you for your comments. Note that the boxes/boundaries used for statistical analysis are only showcased in Figure 1. Those shown in Figure 3 are just general boundaries (which span the full subplots of Figure 1) to point out that these regions exhibit significant unforced variability in $CO_2$ fluxes. The regions in Figures 6–9 are identical to those in Figure 1, but are just zoomed in a bit more on the region. We also point out in the caption of Figure 1 that these boundaries follow the model grid, i.e., that they are confined to 800km offshore zonally along the coarse grid.*

*We updated figure captions to clarify these points. To Figure 1, we added a note that the statistical boxes are confined to 800km "in the E–W direction." To Figure 3, we added "Here, the black boxes outline the general domain of the EBUS in this study but do not coincide with the statistical boundaries shown in Figure 1." Lastly, we renamed Section 2.2 to "Upwelling Regions and Anomalies" to account for the description of anomaly generation in the second half of that section.*

For the regressions onto the climate indices, it should be be more clearly stated what the independent variable and its units are. References to a "1 degree El Nino" or 1 SD of NAO are better than nothing but not really adequate. If NINO3 is defined as a temperature anomaly in K, then a regression coefficient of CO2 flux on this index will have units of mol mËĘ-2 yËĘ-1 KËĘ-1 (e.g., 11/2). Similarly, the NAO has units of SLP. I'm not sure what the units of the NPGO are. But a statement like "The direct regression of DeltaF onto the NPGO results in an anomalous uptake of 0.10 mol mËĘ-2 yrËĘ-1" seems incomplete, because the reader does not know how large an anomaly in the NPGO is required to give rise to X mol mËĘ-2 yrËĘ-1 of CO2 flux

anomaly. The statement (12/10) that stronger winds in the CanCS "leads to the highest relative CO2 flux anomaly of any system" seems misleading because the independent variable for these regressions is different in each case. Maybe there is some basis for making this comparison, but it has not been clearly explained.

*Thank you for this suggestion. We updated all cases where regression results were mentioned to account for the definition of the given mode of climate variability ($K^{-1}$ for Nino3, $\sigma^{-1}$ for NAO, NPGO, PDO). We have removed 12/10 per your suggestion.*

The statistical tests applied are inadequately explained. The discussion of autocorrelation (5/27) appears out of nowhere without context. I agree that autocorrelation is important and you have to correct for it, but up to this point there has been no mention of statistical testing at all. First explain what test you are using to determine whether X is significantly different from 0, and state clearly what physical quantity X represents, then explain the effective sample size. The effective sample size is said to "replace the t-statistic sample size" (5/27), bu there is no mention of t-tests having been conducted; the only test specifically mentioned is the Mann-Kendall test (Table 1).

*Thank you for your suggestion. We added the following description of our correlation/t-test methodology immediately following 5/26:*

*We use a Pearson product-moment correlation for all linear correlations performed in this study (e.g., between area-weighted $CO_2$ flux anomalies and climate indices for each EBUS). Our null hypothesis is that the two time series being compared are uncorrelated, following the Student's t-distribution with a significance level of $\alpha = 0.05$.*

*We also further clarify the description of autocorrelation for the reader, replacing 5/27–28 with (see also response to Reviewer 1):*

*Autocorrelation is prevalent in climate indices such as the NPGO and ENSO (Di Lorenzo and Ohman, 2013), and our annual smoothing further enhances autocorrelation in CalCS and CanCS air-sea $CO_2$ fluxes (see Sections 3.3.1 and 3.3.3). To compensate for this autocorrelation, we replace the t-statistic sample size N with an effective sample size $N_{eff}$, which quantifies the number of statistically independent measurements: . . .*

*Lastly, we add a description of the Mann-Kendall test following 6/2:*

*We use a one-sided Mann-Kendall test to assess significance in trends (e.g., the long-term diffusion of anthropogenic $CO_2$ into EBUS). Our null hypothesis is that the trend is not significantly different from zero, with $\alpha = 0.05$.*

**Model Evaluation:**
I find parts of the discussion of model validation against SOM-FFN confusing. On the one hand, it is the best observational benchmark available, on the other hand, discrepancies are explained away as resulting from errors in the observational data product (6/17-21). I can't make sense of "CO2 fluxes in SOM-FFN are being informed by remote biogeochemical provinces more often than other regions of the ocean". I can guess at what is being stated here, but without a more specific explanation of what

sort of bias it imparts to the data product, I don't think it helps to achieve the task at hand, which is to evaluate how physically realistic the model solution is.

*We removed 6/19–21 ("CO$_2$ fluxes in SOM-FFN are being informed . . . )  to avoid confusion for the reader.*

Describing a model as 'biased' without specifying the nature or sign of the bias (e.g., 6/15, 6/21) is not very useful. The beginning of 3.2 is misleading (CalCS shows very good results, HumCS much less so) and poorly worded. How about "CESM-LENS simulates the pCO2 seasonal cycle well for the Pacific EBUS, with larger error in the Atlantic regions"? (and change "Beginning with the CalCS" to simply "in the CalCS"). Again, this paragraph mixes up model evaluation, analyses of the model solution that do not have any analogue in the observations, and literature review. Again, I think this whole section should be included in the Results, and a clear separation of Results and Discussion attempted.

*Thank you for your suggestion. 6/15 was changed to:*

> *The CO$_2$ flux climatology in the Atlantic systems is more biased in the CESM-LENS, with a tendency for spurious or stronger outgassing than is suggested by the observational product.*

*We feel that the statement immediately following 6/21 describes the nature of the bias, and it would be redundant to mention the outgassing bias in 6/21 directly. We changed*

*the beginning of 3.2 per your suggestions:*

> *CESM-LENS simulates the pCO$_2$ seasonal cycle well for the Pacific EBUS, with larger error in the Atlantic regions. In the CalCS, . . .*

**Terminology:**
"internal" variability in a model simulation is an analogue of "natural" climate variability in the real world. I would prefer that the latter term be used except when the reference is specifically to climate model simulations. I find terms like "have some of the highest internally driven CO2 fluxes globally" very awkward. How about "have some of the highest unforced variability of CO2 flux of any part of the world ocean"?

*Thank you for bringing this important point up. Our use of "internal" over "natural" was intentional in this manuscript. "Natural" variability encompasses both the internal/unforced contribution as well as natural external forcing from volcanic eruptions and the solar cycle. Although we agree that "unforced" is an appropriate alternative to "internal," the authors decided it was best to use "internal" for this study.*

*We have updated the abstract (1/7) to add (unforced) following internal. We have also updated the suggested sentence to "... highest unforced variability ..." for clarity.*

*Further, we have added a very careful description of "internal" and external (both natural and anthropogenic) forcing to the introduction, following 2/18:*

> *Fundamentally, one can differentiate between variability arising from the processes that are purely internal to the climate system, and those that*

*represent "external forcings", i.e., processes that impact the climate sys-
tem from outside. The latter external processes can be further separated
into natural and anthropogenic. The former includes variations induced by
e.g., volcanic eruptions or changes in solar activity, while the latter includes
changes in the concentration of greenhouse gases and other radiatively
active constituents, or human-made changes in albedo. The internal vari-
ability can arise from within a subsystem itself (e.g., baroclinic instabilities
leading to the formation of mesoscale eddies), or from the unforced inter-
action between components of the climate system.*

I think "values" is one of the most overused and abused words in scientific writing.
Search out all occurrences and if possible replace with something specific. For
example, on 10/8 one could replace "r-values" with "correlation coefficients" (see also
9/13-14, 11/22) and on 13/19 "mean values" could be "mean uptake". On 8/11, one
can't even tell what physical quantity is being presented (it is the internal variability
component of the standard deviation of the CO2 flux, but the reader has to go to the
table caption to find this out).

*Thank you for this suggestion. We replaced nearly every occurrence of "value(s)" with
something more specific. We updated both the main text as well as figure labels and
captions in Figs. 5–10.*

There are many locations where "air-sea" could be added before "CO2 flux" in the
interest of clarity (e.g., Figure 10 caption).

*Thank you for this suggestion. We have added "air-sea" as a prefix to "CO$_2$ flux" in the
Figure 10 and Table 1, 2 captions as well as in a few places throughout the text (2/18,
3/16, 3/34, 5/12, Sections 3.1 and 4.2, 8/3)*

**Some details:**

1/21 "Upwelling delivers deep waters with respired nutrients to the surface, fueling primary production and ultimately supporting fisheries that are highly productive with respect to the small surface area they cover" Upwelling delivers waters rich in nutrients to the surface, fueling primary production and ultimately supporting fisheries that are highly productive relative to the small surface area that they cover

*Thank you. We have updated the manuscript to reflect your suggestion.*

2/5 "contributing to the magnitude and determining the direction of air-sea CO2 fluxes" determining the sign and magnitude of the air-sea CO2 flux

*Thank you. We have updated the manuscript to reflect your suggestion.*

2/11 "more efficient biology" greater biological production

*Thank you. We have updated the manuscript to reflect your suggestion.*

2/21 delete "fractional"

*Thank you. We have updated the manuscript to reflect your suggestion.*

3/5 Is it accurate to refer to the NAO as a "decadal-scale oscillation"? I thought it was more like a white noise process with a very flat frequency spectrum.

*Thank you for this clarification. We use the language "decadal-scale" based on section 2 of Hutchings et al. (2009). Note also that the original NAO paper (Hurrell, 1995) suggests that the NAO is a source of low-frequency variability that imparts "large decadal climate variations over the North Atlantic."*

4/20-21 "In contrast ... CO2 fluxes." I would delete this entire sentence.

*Thank you. We have removed this sentence from the manuscript per your suggestion.*

7/13 "These dual peaks are driven by an interchanging importance between thermal and non-thermal effects." These two peaks are driven by an alternating dominance of thermal and non-thermal effects. (see also 12/15 and 12/18)

*Thank you for your suggestion. We have edited 7/13 and 12/15 to reflect these changes.*

7/22 "the BenCS pCO2 seasonal cycle nearly 180 degrees out of phase" This actually true of CanCS as well, although the amplitude is significantly underestimated in the model.

*Thank you for catching this. 7/19 has been modified to read "However, CESM-LENS simulates a damped seasonal cycle that is approximately 180 degrees out of phase for $pCO_2$ in the CanCS ..."*

9/23 change "During a positive NPGO event" to "During the positive NPGO phase"?
(and delete "of the system")

*Thank you. We have updated the manuscript to reflect your suggestion.*

9/24 add "transport" after "DIC"

*Thank you. We have updated the manuscript to reflect your suggestion.*

9/27 "Because the system-wide contributions of SST and sDIC to the anomalous flux nearly balance each other, minor contributions from wind, salinity, sAlk, and freshwater flux push the system in favor of anomalous uptake" I think this is an overinterpretation. It looks to me like the SST contribution is larger than the sDIC, and even if the 4 smaller terms cancelled each other out the net would still be negative.

*Thank you for this comment. Note in Table 2 that the SST contribution is only slightly larger than the sDIC contribution (-0.12 mol $m^{-2}$ $yr^{-1}$ vs. 0.11 mol $m^{-2}$ $yr^{-1}$). Accounting for ensemble spread, some individual ensemble members have sDIC contributions that slightly outweigh SST, so the minor terms are an important contribution toward causing uptake anomalies.*

10/27 change "influencer" to "influence"

*Thank you. We have updated the manuscript to reflect your suggestion.*

11/4 change "are in opposition to one another" to "are of opposite sign"

*Thank you. We have updated the manuscript to reflect your suggestion.*

11/14 change "advected warm waters from the equatorial Pacific" to "warm water advected from the equatorial Pacific"

*Thank you. We have updated the manuscript to reflect your suggestion.*

11/17 change "intensification of wind magnitude" to "increase in wind speed"

*Thank you. We have updated the manuscript to reflect your suggestion.*

11/23 "This encircles the climatological position of the Azores High, the atmospheric subtropical gyre which forces the CanCS." I have never heard the Azores High referred to as an "atmospheric subtropical gyre", although it is a large-scale anticyclone. But I have never heard this terminology before, and it's generally bad practice to take existing terms and assign them new meanings without a compelling reason. I also don't think "encircles" is a good choice of words. How about "represents" (or "indicates" or "coincides with")?

*Thank you for your suggestion. We agree that it is best to avoid assigning new labels. We've updated 11/23 to read "large-scale anticyclone" and have changed out "encircles" to "coincides with." We have also changed reference to "subtropical gyre" with "large-scale anticyclone" where appropriate (12/23)*

11/32 "the linear Taylor approximation aligns exactly with the direct regression" I can't tell what this means.

*Thank you for bringing this up. We have excised this statement to avoid confusion for the reader.*

11/33 "The NAO describes modifications to the intensity of atmospheric gyre circulation between the Azores High and Icelandic Low" The NAO represents fluctuations in the intensity of atmospheric circulation between the Azores High and Icelandic Low

*Thank you. We have updated the manuscript to reflect your suggestion.*

12/21 change "when sDIC and SST are of equal magnitude" to "when the sDIC and SST associated terms are of about equal magnitude"

*Thank you. We have updated the manuscript to reflect your suggestion.*

12/27-30 "The major EBUS ... variability in CO2 fluxes." another truly awful sentence: rewrite or delete

*Thank you for your suggestion. We've excised this sentence as well as the sentences that follow it (12/27–33)*

13/13 change "diffusion" to "mixing"

*Thank you. We have updated the manuscript to reflect your suggestion.*

13/7 delete "to" before "roughly"

*Thank you. We have updated the manuscript to reflect your suggestion.*

13/25 I think this statement requires a data or literature reference.

*Thank you for this suggestion. For clarification, our comments on changes to the $CO_2$ flux seasonal cycle were based on our analysis of the CESM-LENS. We have added statements to this line clarifying that these are results from CESM-LENS to avoid confusion for the reader.*

13/34 "the relative contributions of variables to anomalous CO2 fluxes" the relative contributions of different physical processes to anomalous CO2 fluxes

*Thank you. We have updated the manuscript to reflect your suggestion.*

14/4 "While not observed in our historical modeling study, modifications to modes of climate variability associated with the major EBUS could directly influence the magnitude of internally generated anomalies in CO2 fluxes in the future." I don't see how we know this. Such trends might exist in the ensemble data even if no one has yet attempted to detect them.

*Thank you for your suggestion. We edited the text to be more clear, following 14/1–4 with: "These modifications to modes of climate variability suggested by the literature could directly impact the response of EBUS $CO_2$ flux anomalies to internal variability, thus affecting the conclusions of our study."*

14/9 "we only present the leading mode of climate variability" Similarly, this may be true but I don't think it is demonstrated by the data shown in this paper. The authors simply focus, in each region, on what they EXPECT to be the most important mode; they don't actually test whether this is true.

*Thank you for noticing this. We modified the manuscript to read: "We present the mode of climate variability that has the largest influence on $CO_2$ flux ..."*

14/10-12 "we explain", "we were able to explain" not an appropriate use of first person (I suggest that the wording of all discussions of explained variance in this paragraph be reviewed.)

*Thank you for your suggestion. We updated the manuscript to read "we account for" in both cases.*

14/19 change "a coarse single model ensemble" to "a single coarse-resolution model"

*Thank you. We have updated the manuscript to reflect your suggestion.*

14/24 "do not directly resolve the coastal upwelling process which induces vigorous outgassing within the first O(10km) of the coastline" do not resolve the coastal up-welling that induces vigorous outgassing within the first ~10 km of the coast.

*Thank you. We have updated the manuscript to reflect your suggestion. Note that we used ~50km which is a more accurate depiction of the length scale of outgassing if*

*one is not to use O().*

14/28-29 I agree with this sentiment, but you have to get the boundary conditions for the downscaling model from global models. So if those models have huge biases in the positions of major transition zones, it's not clear that having resolution within a regional domain is going to do any good. This is a problem in the northwest Atlantic as well, as coarse resolution models have large and persistent biases in the location of the Gulf Stream separation from the coast.

*Thank you for your comments. We agree that one requires boundary conditions from global models to downscale to higher resolution. However, Machu et al. (2015) show significant improvement in the physics and biogeochemistry of the Benguela Current through dynamical downscaling (i.e., even when inheriting biases through the coarse boundary conditions). Further, other techniques can help to ensure that the high resolution regional model has less bias than the coarse global model. For instance, Small et al. (2015) reduce the Benguela warm bias by also increasing atmospheric resolution and adjusting alongshore wind stress curl to be more realistic for the system. (Manuscript unchanged in response to comment)*

Figure 5 contains a great deal of information, and the caption could be a bit clearer. Violin plots may not be familiar to some readers, and exactly what is shown in the right hand panels could be spelled out. Similarly, the caption to Table 2 could contain a great deal more detail.

*Thank you for your suggestion. The following has been added to the end of Figure 5's caption:*

*The interior of the violin plot displays a box plot with the ensemble mean denoted as a white dot. Shading around the box plot reflects the ensemble distribution of correlations, which are mirrored on either side.*

*The caption for Table 2 was updated to:*

*Estimated contributions of individual terms to air-sea $CO_2$ flux anomalies, $\Delta F$, in response to a mode of climate variability using Equation 4. Each row under the Individual Terms header depicts the ensemble mean contribution and ensemble spread for Figures 6–9. The column "CalCS–PDOn" reflects results from the nearshore box in the CalCS in Figure 7, and "CalCS–PDOo" the offshore box. $\Sigma$ is the sum of all contributing first-order terms (i.e., all rows under Individual Terms or the right hand side of Equation 4). $\Delta F$ is the direct regression of $CO_2$ flux anomalies onto the specified mode of climate variability (i.e., the left hand side of Equation 4).*

*We also included units in Table 2 and Table 3 to reflect that these are responses to 1 unit of the given mode of climate variability (e.g., mol m$^{-2}$ yr$^{-1}$ $\sigma^{-1}$)*

The BenCS has larger physical biases in CESM-LENS than all other EBUS. Its SST bias is in excess of $7^o$C with the nominal

15  $1^o$ atmospheric resolution, compared to less than a $1^o$C bias in the CalCS and CanCS, and a 1–$3^o$C bias in the HumCS (pers. comm. with RJ Small, 2018). Further, the BenCS only improves to a $5^o$C bias at $0.5^o$ atmospheric resolution (Gent et al., 2010). This bias is likely driven by the fact that the Angola-Benguela Front is simulated too far south, in addition to deficiencies in upwelling and meridional transport that are caused by unrealistic alongshore wind stress structure (Small et al., 2015). Because these deficiencies are specific to the BenCS, we will only discuss its representation of the $pCO_2$ seasonal cycle in Section

20  3.23.1.2, and its internal variability in $CO_2$ fluxes in Section 4.13.2, but will not perform a full analysis on its connections to larger-scale climate variability.

**3.2**

**3.1.1 $p$CO$_2$ Seasonal Cycle**

CESM-LENS simulates the $pCO_2$ seasonal cycle for the EBUS with similar accuracy to its depiction of the mean state – the

25  Pacific systems are generally well-modeled, while larger biases exist well for the Pacific EBUS, with larger error in the Atlantic regions. Beginning with EBUS. In the CalCS, CESM-LENS nearly perfectly matches the SOM-FFN in both amplitude and phase (Figure 2a). The system exhibits its maximum $pCO_2$ (and thus $CO_2$ outgassing) in August, and its minimum $pCO_2$ (and thus $CO_2$ uptake) in April. We further decomposed the model seasonal cycle into its thermal component (driven by the seasonality of SST) and its non-thermal component (driven by the seasonality of factors such as DIC, ALK, and salinity)

30  following Takahashi et al. (2002). This decomposition suggests that the phase of the CalCS seasonal cycle is determined by thermal (solubility) effects, with its amplitude modulated by non-thermal factors (Figure 2a). These non-thermal factors are almost entirely driven by the seasonal cycle of DIC (not shown), which is characterized by photosynthetic uptake of $CO_2$ in the summer and fall, coinciding with upwelling season. These dynamics are supported by the high-resolution modeling study of

 In the HumCS, both CESM-LENS and SOM-FFN suggest a dual peak in the seasonal cycle of $p$CO$_2$, although the model and observational product slightly disagree in phase and amplitude (Figure 2b). These  two peaks are driven by an  alternating dominance of thermal and non-thermal effects. During the austral summer/fall, warm temperatures lead to enhanced $p$CO$_2$ that is slightly compensated for by increased biological activity, similar to the singular peak of the CalCS. In the austral winter/spring, intense upwelling of DIC-enriched waters and reduced biological activity that is slightly compensated for by cooler waters leads to a secondary $p$CO$_2$ peak (Kämpf and Chapman, 2016). The SOM-FFN suggests that the CanCS behaves similarly to the CalCS, with a single $p$CO$_2$ peak in late summer/fall that is in phase with the seasonal cycle of SST (Figure 2c). However, CESM-LENS simulates  a damped seasonal cycle that is approximately 180 degrees out of phase for $p$CO$_2$ in the CanCS that results from a delicate balance between thermal and non-thermal effects of similar magnitudes. Despite the thermal and non-thermal effects being in the proper phase for a northern hemisphere system, the SST seasonal cycle is too weak and the DIC seasonal cycle too strong in the CanCS in CESM-LENS. Lastly, we find that CESM-LENS simulates the BenCS $p$CO$_2$ seasonal cycle nearly 180 degrees out of phase with the SOM-FFN representation (Figure 2d). Similar to the CanCS, the thermal and non-thermal effects are in the proper phase. However, there is a large bias in the magnitude of the DIC seasonal cycle, which overwhelms the thermal seasonality and drives the $p$CO$_2$ seasonal cycle entirely out of phase. ~~This bias in the $p$CO$_2$ seasonal cycle further illuminates deficiencies in CESM1 in simulating the BenCS. Note that for all the EBUS, the dominance of the thermal effects is likely due to the fact that our area-weighted region encompasses a large domain that is driven by gyre-scale dynamics and downwelling. We would expect non-thermal factors to play an important role in dictating $p$CO$_2$ seasonality nearshore in a high-resolution simulation, such as in Turi et al. (2014), their Figure 9b.~~

**4**

[revised manuscript text omitted]
., 1999). ~~The major EBUS are often lumped together in studies due to their similarities – they are all characterized by their presence on the eastern flank of subtropical gyres, their Ekman dynamics associated with this positioning which leads to coastal and curl-driven upwelling, their productive fisheries, and in the case of our study, their high variability in CO$_2$ fluxes. However, we show in this study that their position in terms of latitude and ocean basin as well as their coastal geometry leads to unique physical and biogeochemical responses to climate variability. In particular, despite variations in sDIC being a leading contributor to CO$_2$ flux anomalies, the drivers of these sDIC anomalies differ between EBUS.~~

[revised manuscript text omitted]

D. C. E. Bakker, B. Pfeil, K. M. O'Brien, K. I. Currie, S. D. Jones, C. S. Landa, S. K. Lauvset, N. Metzl, D. R. Munro, S.-I. Nakaoka, A. Olsen, D. Pierrot, S. Saito, K. Smith, C. Sweeney, T. Takahashi, C. Wada, R. Wanninkhof, S. R. Alin, M. Becker, R. G. J. Bellerby, A. V. Borges, J. Boutin, Y. Bozec, E. Burger, W.-J. Cai, R. D. Castle, C. E. Cosca, M. D. DeGrandpre, M. Donnelly, G. Eischeid, R. A. Feely, T. Gkritzalis, M. González-Dávila, C. Goyet, A. Guillot, N. J. Hardman-Mountford, J. Hauck, M. Hoppema, M. P. Humphreys, C. W. Hunt, J. S. P. Ibánhez, T. Ichikawa, M. Ishii, L. W. Juranek, V. Kitidis, A. Körtzinger, U. K. Koffi, A. Kozyr, A. Kuwata, N. Lefèvre, C. Lo Monaco, A. Manke, P. Marrec, J. T. Mathis, F. J. Millero, N. Monacci, P. M. S. Monteiro, A. Murata, T. Newberger, Y. Nojiri, I. Nonaka, A. M. Omar, T. Ono, X. A. Padín, G. Rehder, A. Rutgersson, C. L. Sabine, J. Salisbury, J. M. Santana-Casiano, D. Sasano, U. Schuster, R. Sieger, I. Skjelvan, T. Steinhoff, K. Sullivan, S. C. Sutherland, A. Sutton, K. Tadokoro, M. Telszewski, H. Thomas, B. Tilbrook, S. van Heuven, D. Vandemark, D. W. Wallace, and R. Woosley. Surface Ocean CO2 Atlas (SOCAT) V4, 2016.

A. Bakun, B. A. Black, S. J. Bograd, M. García-Reyes, A. J. Miller, R. R. Rykaczewski, and W. J. Sydeman. Anticipated effects of climate change on coastal upwelling ecosystems. *Current Climate Change Reports*, 1(2):85–93, Jun 2015. ISSN 2198-6061. https://doi.org/10.1007/s40641-015-0008-4.

R. T. Barber and F. P. Chavez. Biological consequences of El Niño. *Science*, 222(4629):1203–1210, 1983. ISSN 0036-8075. https://doi.org/10.1126/science.222.4629.1203.

R. T. Barber and F. P. Chávez. Ocean variability in relation to living resources during the 1982–83 El Niño. *Nature*, 319:279 EP –, 01 1986.

A. V. Borges and M. Frankignoulle. Distribution of surface carbon dioxide and air-sea exchange in the upwelling system off the galician coast. *Global Biogeochemical Cycles*, 16(2), 2002.

M. F. Borges, A. M. P. Santos, N. Crato, H. Mendes, and B. Mota. Sardine regime shifts off Portugal: a time series analysis of catches and wind conditions. *Scientia Marina*, 67(S1):235–244, Apr. 2003. ISSN 1886-8134, 0214-8358. https://doi.org/10.3989/scimar.2003.67s1235.

A. Boyd, J. Salat, and M. Masó. The seasonal intrusion of relatively saline water on the shelf off northern and central Namibia. *South African Journal of Marine Science*, 5(1):107–120, 1987.

R. X. Brady, M. A. Alexander, N. S. Lovenduski, and R. R. Rykaczewski. Emergent anthropogenic trends in California Current upwelling. *Geophysical Research Letters*, 44(10):5044–5052, 2017.

C. S. Bretherton, M. Widmann, V. P. Dymnikov, J. M. Wallace, and I. Bladé. The effective number of spatial degrees of freedom of a time-varying field. *Journal of Climate*, 12(7):1990–2009, 1999. https://doi.org/10.1175/1520-0442(1999)012<1990:TENOSD>2.0.CO;2.

W. Cai, S. Borlace, M. Lengaigne, P. Van Rensch, M. Collins, G. Vecchi, A. Timmermann, A. Santoso, M. J. McPhaden, L. Wu, et al. Increasing frequency of extreme El Niño events due to greenhouse warming. *Nature climate change*, 4(2):111, 2014.

W. Cai, G. Wang, A. Santoso, M. J. McPhaden, L. Wu, F.-F. Jin, A. Timmermann, M. Collins, G. Vecchi, M. Lengaigne, et al. Increased frequency of extreme La Niña events under greenhouse warming. *Nature Climate Change*, 5(2):132–137, 2015.

W.-J. Cai, M. Dai, and Y. Wang. Air-sea exchange of carbon dioxide in ocean margins: A province-based synthesis. *Geophysical Research Letters*, 33(12), 2006. https://doi.org/10.1029/2006GL026219.

Antonietta Capotondi, Andrew T Wittenberg, Matthew Newman, Emanuele Di Lorenzo, Jin-Yi Yu, Pascale Braconnot, Julia Cole, Boris Dewitte, Benjamin Giese, Eric Guilyardi, et al. Understanding ENSO diversity. *Bulletin of the American Meteorological Society*, 96(6): 921–938, 2015.

F. Chan, J. A. Barth, C. A. Blanchette, R. H. Byrne, F. Chavez, O. Cheriton, R. A. Feely, G. Friederich, B. Gaylord, T. Gouhier, S. Hacker, T. Hill, G. Hofmann, M. A. McManus, B. A. Menge, K. J. Nielsen, A. Russell, E. Sanford, J. Sevadjian, and L. Washburn. Persistent spatial structuring of coastal ocean acidification in the California Current System. *Scientific Reports*, 7(1):2526, 2017. https://doi.org/10.1038/s41598-017-02777-y.

F. P. Chavez and M. Messié. A comparison of Eastern Boundary Upwelling Ecosystems. *Progress in Oceanography*, 83(1):80–96, 2009. https://doi.org/https://doi.org/10.1016/j.pocean.2009.07.032.

5   F. P. Chavez, J. T. Pennington, C. G. Castro, J. P. Ryan, R. P. Michisaki, B. Schlining, P. Walz, K. R. Buck, A. McFadyen, and C. A. Collins. Biological and chemical consequences of the 1997–1998 El Niño in central california waters. *Progress in Oceanography*, 54(1):205–232, 2002. https://doi.org/https://doi.org/10.1016/S0079-6611(02)00050-2.

FP Chavez, PG Strutton, GE Friederich, RA Feely, GC Feldman, DG Foley, and MJ McPhaden. Biological and chemical response of the equatorial Pacific Ocean to the 1997-98 El Niño. *Science*, 286(5447):2126–2131, 1999.

10   D. B. Chelton, P. A. Bernal, and J. A. McGowan. Large-scale interannual physical and biological interaction in the California Current. *Journal of Marine Research*, 40(4):1095–1125, 1982.

F. Chenillat, P. Rivière, X. Capet, E. Di Lorenzo, and B. Blanke. North Pacific Gyre Oscillation modulates seasonal timing and ecosystem functioning in the California Current upwelling system. *Geophysical Research Letters*, 39(1), 2012. https://doi.org/10.1029/2011GL049966.

15   K. Chhak and E. Di Lorenzo. Decadal variations in the California Current upwelling cells. *Geophysical Research Letters*, 34(14), 2007. https://doi.org/10.1029/2007GL030203.

F. Colas, X. Capet, J. McWilliams, and A. Shchepetkin. 1997–1998 El Niño off Peru: A numerical study. *Progress in Oceanography*, 79 (2-4):138–155, 2008.

T. E. Cropper, E. Hanna, and G. R. Bigg. Spatial and temporal seasonal trends in coastal upwelling off Northwest Africa, 1981–2012. *Deep
20   Sea Research Part I: Oceanographic Research Papers*, 86:94–111, 2014. https://doi.org/https://doi.org/10.1016/j.dsr.2014.01.007.

M. D. DeGrandpre, T. R. Hammar, and C. D. Wirick. Short-term pCO2 and O2 dynamics in California coastal waters. *Deep Sea Research Part II: Topical Studies in Oceanography*, 45(8):1557–1575, 1998. https://doi.org/https://doi.org/10.1016/S0967-0645(98)80006-4.

C. Deser, I. R. Simpson, K. A. McKinnon, and A. S. Phillips. The Northern Hemisphere extratropical atmospheric circulation response to ENSO: How well do we know it and how do we evaluate models accordingly? *Journal of Climate*, 30(13):5059–5082, 2017.

25   C. Deser, I. R. Simpson, A. S. Phillips, and K. A. McKinnon. How well do we know ENSO's climate impacts over North America, and how do we evaluate models accordingly? *Journal of Climate*, 2018.

E. Di Lorenzo and N. Mantua. Multi-year persistence of the 2014/15 North Pacific marine heatwave. *Nature Climate Change*, 6(11): 1042–1047, 2016.

E. Di Lorenzo, A. J. Miller, N. Schneider, and J. C. McWilliams. The warming of the California Current System: Dynamics and ecosystem
30   implications. *Journal of Physical Oceanography*, 35(3):336–362, 2005.

E. Di Lorenzo, N. Schneider, K. M. Cobb, P. J. S. Franks, K. Chhak, A. J. Miller, J. C. McWilliams, S. J. Bograd, H. Arango, E. Curchitser, T. M. Powell, and P. Rivière. North Pacific Gyre Oscillation links ocean climate and ecosystem change. *Geophysical Research Letters*, 35(8), 2008. https://doi.org/10.1029/2007GL032838.

E. Di Lorenzo, J. Fiechter, N. Schneider, A. Bracco, A. J. Miller, P. J. S. Franks, S. J. Bograd, A. M. Moore, A. C. Thomas, W. Crawford,
35   A. Peña, and A. J. Hermann. Nutrient and salinity decadal variations in the central and eastern North Pacific. *Geophysical Research Letters*, 36(14), 2009. https://doi.org/10.1029/2009GL038261.

Emanuele Di Lorenzo and Mark D Ohman. A double-integration hypothesis to explain ocean ecosystem response to climate forcing. *Proceedings of the National Academy of Sciences*, 110(7):2496–2499, 2013.

D. Enfield. Thermally driven wind variability in the planetary boundary layer above Lima, Peru. *Journal of Geophysical Research: Oceans*, 86(C3):2005–2016, 1981.

R. Escribano, G. Daneri, L. Farías, V. A. Gallardo, H. E. González, D. Gutiérrez, C. B. Lange, C. E. Morales, O. Pizarro, O. Ulloa, and M. Braun. Biological and chemical consequences of the 1997–1998 El Niño in the Chilean coastal upwelling system: a synthesis. *Deep Sea Research Part II: Topical Studies in Oceanography*, 51(20):2389–2411, 2004. https://doi.org/https://doi.org/10.1016/j.dsr2.2004.08.011.

W. Evans, B. Hales, and P. G. Strutton. Seasonal cycle of surface ocean pCO2 on the Oregon shelf. *Journal of Geophysical Research: Oceans*, 116(C5), 2011. https://doi.org/10.1029/2010JC006625.

RA Feely, T Takahashi, R Wanninkhof, MJ McPhaden, CE Cosca, SC Sutherland, and Mary-Elena Carr. Decadal variability of the air-sea co2 fluxes in the equatorial pacific ocean. *Journal of Geophysical Research: Oceans*, 111(C8), 2006.

Richard A Feely, Rik Wanninkhof, Taro Takahashi, and Pieter Tans. Influence of El Niño on the equatorial Pacific contribution to atmospheric CO2 accumulation. *Nature*, 398(6728):597, 1999.

Jerome Fiechter, Enrique N Curchitser, Christopher A Edwards, Fei Chai, Nicole L Goebel, and Francisco P Chavez. Air-sea CO2 fluxes in the California Current: Impacts of model resolution and coastal topography. *Global Biogeochemical Cycles*, 28(4):371–385, 2014.

G. E. Friederich, P. M. Walz, M. G. Burczynski, and F. P. Chavez. Inorganic carbon in the central California upwelling system during the 1997–1999 El Niño–La Niña event. *Progress in Oceanography*, 54(1):185–203, 2002. https://doi.org/https://doi.org/10.1016/S0079-6611(02)00049-6.

M. Frischknecht, M. Münnich, and N. Gruber. Remote versus local influence of ENSO on the California Current System. *Journal of Geophysical Research: Oceans*, 120(2):1353–1374, 2015. https://doi.org/10.1002/2014JC010531.

M. Frischknecht, M. Münnich, and N. Gruber. Local atmospheric forcing driving an unexpected California Current System response during the 2015–2016 El Niño. *Geophysical Research Letters*, 44(1):304–311, 2017. https://doi.org/10.1002/2016GL071316.

M. García-Reyes, W. J. Sydeman, D. S. Schoeman, R. R. Rykaczewski, B. A. Black, A. J. Smit, and S. J. Bograd. Under pressure: Climate change, upwelling, and Eastern Boundary Upwelling Ecosystems. *Frontiers in Marine Science*, 2:109, 2015.

P. R. Gent, S. G. Yeager, R. B. Neale, S. Levis, and D. A. Bailey. Improvements in a half degree atmosphere/land version of the CCSM. *Climate Dynamics*, 34(6):819–833, 2010.

M. González-Dávila, J. M. Santana-Casiano, and I. R. Ucha. Seasonal variability of fCO2 in the Angola-Benguela region. *Progress in Oceanography*, 83(1-4):124–133, 2009.

L. Gregor and P. Monteiro. Is the southern Benguela a significant regional sink of CO2? *South African Journal of Science*, 109(5-6):01–05, 2013.

N. Gruber. Carbon at the coastal interface. *Nature; London*, 517(7533):148–149, Jan. 2015. ISSN 00280836.

N. Gruber, C. D. Keeling, and N. R. Bates. Interannual variability in the North Atlantic Ocean carbon sink. *Science*, 298(5602):2374–2378, 2002. ISSN 0036-8075. https://doi.org/10.1126/science.1077077.

B. Hales, T. Takahashi, and L. Bandstra. Atmospheric CO2 uptake by a coastal upwelling system. *Global Biogeochemical Cycles*, 19(1), 2005. https://doi.org/10.1029/2004GB002295.

J. Hauck. Unsteady seasons in the sea. *Nature Climate Change*, 8(2):97–98, 2018. https://doi.org/10.1038/s41558-018-0069-1.

C. Hauri, N. Gruber, G.-K. Plattner, S. Alin, R. A. Feely, B. Hales, and P. A. Wheeler. Ocean Acidification in the California Current System. *Oceanography*, 22(4):60–71, Dec. 2009.

Stephanie A Henson, Claudie Beaulieu, and Richard Lampitt. Observing climate change trends in ocean biogeochemistry: when and where. *Global change biology*, 22(4):1561–1571, 2016.

James W Hurrell, Yochanan Kushnir, and Martin Visbeck. The North Atlantic Oscillation. *Science*, 291(5504):603–605, 2001.

James W Hurrell, Marika M Holland, Peter R Gent, Steven Ghan, Jennifer E Kay, Paul J Kushner, J-F Lamarque, William G Large, D Lawrence, Keith Lindsay, et al. The Community Earth System Model: A framework for collaborative research. *Bulletin of the American Meteorological Society*, 94(9):1339–1360, 2013.

5  L Hutchings, CD Van der Lingen, LJ Shannon, RJM Crawford, HMS Verheye, CH Bartholomae, AK Van der Plas, D Louw, A Kreiner, M Ostrowski, et al. The Benguela Current: An ecosystem of four components. *Progress in Oceanography*, 83(1-4):15–32, 2009.

A. Huyer, R. L. Smith, and T. Paluszkiewicz. Coastal upwelling off Peru during normal and El Niño times, 1981–1984. *Journal of Geophysical Research: Oceans*, 92(C13):14297–14307, 1987.

M. G. Jacox, S. J. Bograd, E. L. Hazen, and J. Fiechter. Sensitivity of the California Current nutrient supply to wind, heat, and remote ocean
10     forcing. *Geophysical Research Letters*, 42(14):5950–5957, 2015.

Jochen Kämpf and Piers Chapman. *Upwelling Systems of the World*. Springer, 2016.

J. E. Kay, C. Deser, A. Phillips, A. Mai, C. Hannay, G. Strand, J. M. Arblaster, S. C. Bates, G. Danabasoglu, J. Edwards, M. Holland, P. Kushner, J.-F. Lamarque, D. Lawrence, K. Lindsay, A. Middleton, E. Munoz, R. Neale, K. Oleson, L. Polvani, and M. Vertenstein. The Community Earth System Model (CESM) Large Ensemble Project: A community resource for studying climate change in the presence of
15     internal climate variability. *Bulletin of the American Meteorological Society*, 96(8):1333–1349, 2015. https://doi.org/10.1175/BAMS-D-13-00255.1.

A. W. King, L. Dilling, G. P. Zimmerman, D. M. Fairman, R. A. Houghton, G. Marland, A. Z. Rose, and T. J. Wilbanks. *The first state of the carbon cycle report (SOCCR): The North American carbon budget and implications for the global carbon cycle.* U.S. Climate Change Science Program, Washington, 2007.

20  S. I. Kuzmina, L. Bengtsson, O. M. Johannessen, H. Drange, L. P. Bobylev, and M. W. Miles. The North Atlantic Oscillation and greenhouse-gas forcing. *Geophysical Research Letters*, 32(4), 2005.

L. Kwiatkowski and J. C. Orr. Diverging seasonal extremes for ocean acidification during the twenty-first century. *Nature Climate Change*, 8(2):141–145, 2018. https://doi.org/10.1038/s41558-017-0054-0.

P. Landschützer, N. Gruber, D. C. E. Bakker, U. Schuster, S. Nakaoka, M. R. Payne, T. P. Sasse, and J. Zeng. A neural network-based
25     estimate of the seasonal to inter-annual variability of the Atlantic Ocean carbon sink. *Biogeosciences*, 10(11):7793–7815, Nov. 2013. ISSN 1726-4189. https://doi.org/10.5194/bg-10-7793-2013.

P Landschützer, N Gruber, DCE Bakker, and U Schuster. Recent variability of the global ocean carbon sink. *Global Biogeochemical Cycles*, 28(9):927–949, 2014.

P. Landschützer, N. Gruber, and D. C. E. Bakker. Decadal variations and trends of the global ocean carbon sink. *Global Biogeochemical*
30     *Cycles*, 30(10):1396–1417, 2016. ISSN 1944-9224. https://doi.org/10.1002/2015GB005359.

P. Landschützer, N. Gruber, and D. C. E. Bakker. An updated observation-based global monthly gridded sea surface pCO2 and air-sea CO2 flux product from 1982 through 2015 and its monthly climatology (ncei accession 0160558). Version 2.2. NOAA National Centers for Environmental Information. Dataset., July 2017.

P. Landschützer, N. Gruber, D. C. E. Bakker, I. Stemmler, and K. D. Six. Strengthening seasonal marine CO2 variations due to increasing
35     atmospheric CO2. *Nature Climate Change*, Jan. 2018. ISSN 1758-678X, 1758-6798. https://doi.org/10.1038/s41558-017-0057-x.

G. G. Laruelle, H. H. Dürr, C. P. Slomp, and A. V. Borges. Evaluation of sinks and sources of CO2 in the global coastal ocean using a spatially-explicit typology of estuaries and continental shelves. *Geophysical Research Letters*, 37(15), Aug. 2010. ISSN 00948276. https://doi.org/10.1029/2010GL043691.

G. G. Laruelle, R. Lauerwald, B. Pfeil, and P. Regnier. Regionalized global budget of the CO2 exchange at the air-water interface in continental shelf seas. *Global Biogeochem. Cycles*, 28(11):2014GB004832, Nov. 2014. ISSN 1944-9224. https://doi.org/10.1002/2014GB004832.

G. G. Laruelle, P. Landschützer, N. Gruber, J.-L. Tison, B. Delille, and P. Regnier. Global high-resolution monthly pCO2 climatology for the coastal ocean derived from neural network interpolation. *Biogeosciences*, 14(19):4545, 2017.

5    A. Leinweber, N. Gruber, H. Frenzel, G. E. Friederich, and F. P. Chavez. Diurnal carbon cycling in the surface ocean and lower atmosphere of Santa Monica Bay, California. *Geophysical Research Letters*, 36(8), Apr. 2009. ISSN 0094-8276. https://doi.org/10.1029/2008GL037018.

Keith Lindsay, Gordon B Bonan, Scott C Doney, Forrest M Hoffman, David M Lawrence, Matthew C Long, Natalie M Mahowald, J Keith Moore, James T Randerson, and Peter E Thornton. Preindustrial-control and twentieth-century carbon cycle experiments with the Earth System Model CESM1 (BGC). *Journal of Climate*, 27(24):8981–9005, 2014.

10    N. Lovenduski, M. Long, and K. Lindsay. Natural variability in the surface ocean carbonate ion concentration. *Biogeosciences*, 12(21): 6321–6335, 2015.

N. S. Lovenduski and N. Gruber. Impact of the Southern Annular Mode on Southern Ocean circulation and biology. *Geophys. Res. Lett.*, 32 (11):L11603, June 2005. ISSN 1944-8007. https://doi.org/10.1029/2005GL022727.

N. S. Lovenduski, N. Gruber, S. C. Doney, and I. D. Lima. Enhanced CO2 outgassing in the Southern Ocean from a positive phase of the

15    Southern Annular Mode. *Global Biogeochem. Cycles*, 21(2), June 2007. ISSN 1944-9224. https://doi.org/10.1029/2006GB002900.

N. S. Lovenduski, G. A. McKinley, A. R. Fay, K. Lindsay, and M. C. Long. Partitioning uncertainty in ocean carbon uptake projections: Internal variability, emission scenario, and model structure. *Global Biogeochemical Cycles*, 30(9):1276–1287, 2016.

R. Lynn and S. Bograd. Dynamic evolution of the 1997–1999 El Niño–La Niña cycle in the southern California Current System. *Progress in Oceanography*, 54(1-4):59–75, July 2002. ISSN 00796611. https://doi.org/10.1016/S0079-6611(02)00043-5.

20    N. J. Mantua and S. R. Hare. The Pacific Decadal Oscillation. *Journal of oceanography*, 58(1):35–44, 2002.

N. J. Mantua, S. R. Hare, Y. Zhang, J. M. Wallace, and R. C. Francis. A Pacific Interdecadal Climate Oscillation with impacts on salmon production. *Bulletin of the American Meteorological Society*, 78(6):1069–1079, June 1997. ISSN 0003-0007. https://doi.org/10.1175/1520-0477(1997)078<1069:APICOW>2.0.CO;2.

G. A. McKinley, T. Takahashi, E. Buitenhuis, F. Chai, J. R. Christian, S. C. Doney, M.-S. Jiang, K. Lindsay, J. K. Moore, C. Le Quere, et al.

25    North Pacific carbon cycle response to climate variability on seasonal to decadal timescales. *Journal of Geophysical Research: Oceans*, 111(C7), 2006.

R. Mogollón and P. H. Calil. On the effects of ENSO on ocean biogeochemistry in the Northern Humboldt Current System (NHCS): A modeling study. *Journal of Marine Systems*, 172:137–159, 2017.

I. Montes, W. Schneider, F. Colas, B. Blanke, and V. Echevin. Subsurface connections in the eastern tropical Pacific during La Niña 1999–

30    2001 and El Niño 2002–2003. *Journal of Geophysical Research: Oceans*, 116(C12), 2011.

J Keith Moore, Keith Lindsay, Scott C Doney, Matthew C Long, and Kazuhiro Misumi. Marine ecosystem dynamics and biogeochemical cycling in the Community Earth System Model [CESM1 (BGC)]: Comparison of the 1990s with the 2090s under the RCP4. 5 and RCP8. 5 scenarios. *Journal of Climate*, 26(23):9291–9312, 2013.

N. Narayan, A. Paul, S. Mulitza, and M. Schulz. Trends in coastal upwelling intensity during the late 20th century. *Ocean Science*, 6(3):815,

35    2010.

V. Oerder, F. Colas, V. Echevin, F. Codron, J. Tam, and A. Belmadani. Peru-Chile upwelling dynamics under climate change. *Journal of Geophysical Research: Oceans*, 120(2):1152–1172, 2015.

Daniel Pauly and Villy Christensen. Primary production required to sustain global fisheries. *Nature*, 374(6519):255–257, 1995.

Adam S Phillips, Clara Deser, and John Fasullo. Evaluating modes of variability in climate models. *Eos, Transactions American Geophysical Union*, 95(49):453–455, 2014.

M. Pozo Buil and E. Di Lorenzo. Decadal dynamics and predictability of oxygen and subsurface tracers in the California Current System. *Geophysical Research Letters*, 44(9):4204–4213, 2017.

5   C. Reason, P. Florenchie, M. Rouault, and J. Veitch. Influences of large scale climate modes and Agulhas system variability on the BCLME region. In *Large Marine Ecosystems*, volume 14, pages 223–238. Elsevier, 2006. ISBN 978-0-444-52759-2. https://doi.org/10.1016/S1570-0461(06)80015-7.

R. R. Rykaczewski and J. P. Dunne. Enhanced nutrient supply to the California Current Ecosystem with global warming and increased stratification in an Earth System Model. *Geophysical Research Letters*, 37(21), 2010.

10   R. R. Rykaczewski, J. P. Dunne, W. J. Sydeman, M. García-Reyes, B. A. Black, and S. J. Bograd. Poleward displacement of coastal upwelling-favorable winds in the ocean's eastern boundary currents through the 21st century. *Geophysical Research Letters*, 42(15):6424–6431, 2015.

J. H. Ryther. Photosynthesis and fish production in the sea. *Science*, 166(3901):72–76, 1969.

J. M. Santana-Casiano, M. González-Dávila, M.-J. Rueda, O. Llinás, and E.-F. González-Dávila. The interannual variability of oceanic CO2

15   parameters in the northeast Atlantic subtropical gyre at the ESTOC site. *Global Biogeochemical Cycles*, 21(1), 2007.

L. V. Shannon, A. J. Boyd, G. B. Brundrit, and J. Taunton-Clark. On the existence of an El Niño-type phenomenon in the Benguela System. *Journal of Marine Research*, 44(3):495–520, August 1986. https://doi.org/10.1357/0022240867884031305.

R. J. Small, E. Curchitser, K. Hedstrom, B. Kauffman, and W. G. Large. The Benguela upwelling system: Quantifying the sensitivity to resolution and coastal wind representation in a global climate model. *Journal of Climate*, 28(23):9409–9432, 2015.

20   R Smith, P Jones, B Briegleb, F Bryan, G Danabasoglu, J Dennis, J Dukowicz, C Eden, B Fox-Kemper, P Gent, et al. The Parallel Ocean Program (POP) reference manual ocean component of the Community Climate System Model (CCSM) and Community Earth System Model (CESM). *Rep. LAUR-01853*, 141:1–140, 2010.

P. T. Strub, J. Mesias, V. Montecino, J. Rutllant, and S. Marchant. Coastal ocean circulation off western South America. In *The Sea*, volume 11. John Wiley, New York, 1998.

25   W. Sydeman, M. García-Reyes, D. Schoeman, R. Rykaczewski, S. Thompson, B. Black, and S. Bograd. Climate change and wind intensification in coastal upwelling ecosystems. *Science*, 345(6192):77–80, 2014.

W. J. Sydeman, J. A. Santora, S. A. Thompson, B. Marinovic, and E. D. Lorenzo. Increasing variance in North Pacific climate relates to unprecedented ecosystem variability off California. *Global Change Biology*, 19(6):1662–1675, 2013.

T. Takahashi, S. C. Sutherland, R. Wanninkhof, C. Sweeney, R. A. Feely, D. W. Chipman, B. Hales, G. Friederich, F. Chavez, C. Sabine,

30   A. Watson, D. C. E. Bakker, U. Schuster, N. Metzl, H. Yoshikawa-Inoue, M. Ishii, T. Midorikawa, Y. Nojiri, A. Körtzinger, T. Steinhoff, M. Hoppema, J. Olafsson, T. S. Arnarson, B. Tilbrook, T. Johannessen, A. Olsen, R. Bellerby, C. S. Wong, B. Delille, N. R. Bates, and H. J. W. de Baar. Climatological mean and decadal change in surface ocean pCO2, and net sea–air CO2 flux over the global oceans. *Deep Sea Research Part II: Topical Studies in Oceanography*, 56(8):554–577, Apr. 2009. ISSN 0967-0645. https://doi.org/10.1016/j.dsr2.2008.12.009.

35   Taro Takahashi, Stewart C Sutherland, Richard A Feely, and Catherine E Cosca. Decadal variation of the surface water pco2 in the western and central equatorial pacific. *Science*, 302(5646):852–856, 2003.

Taro Takahashi, Stewart C. Sutherland, Colm Sweeney, Alain Poisson, Nicolas Metzl, Bronte Tilbrook, Nicolas Bates, Rik Wanninkhof, Richard A. Feely, Christopher Sabine, Jon Olafsson, and Yukihiro Nojiri. Global sea–air CO2 flux based on climatological surface ocean

pCO2, and seasonal biological and temperature effects. *Deep Sea Research Part II: Topical Studies in Oceanography*, 49(9):1601–1622, January 2002. ISSN 0967-0645. https://doi.org/10.1016/S0967-0645(02)00003-6.

D. W. J. Thompson, E. A. Barnes, C. Deser, W. E. Foust, and A. S. Phillips. Quantifying the role of internal climate variability in future climate trends. *Journal of Climate*, 28(16):6443–6456, 2015. https://doi.org/10.1175/JCLI-D-14-00830.1.

5   A. Timmermann, J. Oberhuber, A. Bacher, M. Esch, M. Latif, and E. Roeckner. Increased El Niño frequency in a climate model forced by future greenhouse warming. *Nature*, 398(6729):694, 1999.

Rodrigo Torres, David R Turner, José Rutllant, and Nathalie Lefèvre. Continued CO2 outgassing in an upwelling area off northern Chile during the development phase of El Niño 1997–1998 (July 1997). *Journal of Geophysical Research: Oceans*, 108(C10), 2003.

G. Turi, Z. Lachkar, and N. Gruber. Spatiotemporal variability and drivers of pCO2 and air–sea CO2 fluxes in the California Current System:

10   an eddy-resolving modeling study. *Biogeosciences*, 11(3):671–690, Feb. 2014. ISSN 1726-4189. https://doi.org/10.5194/bg-11-671-2014.

Giuliana Turi, Zouhair Lachkar, Nicolas Gruber, and Martin Münnich. Climatic modulation of recent trends in ocean acidification in the California Current System. *Environmental Research Letters*, 11(1):014007, 2016.

G. Turi, M. A. Alexander, N. S. Lovenduski, A. Capotondi, J. D. Scott, C. A. Stock, J. P. Dunne, J. John, and M. G. Jacox. Response of O2 and pH to ENSO in the California Current System in a high resolution global climate model. *Ocean Sciences Discussion*, Aug. 2017.

[revised manuscript text omitted]
$ | $-0.01 \pm 0.0$ | $0.01 \pm 0.0$ | $0.0 \pm 0.0$ | $0.06 \pm 0.01$ | $0.05 \pm 0.01$ |
| $\frac{\partial F}{\partial T}\Delta T$ | $-0.12 \pm 0.02$ | $0.28 \pm 0.03$ | $0.03 \pm 0.0$ | $0.29 \pm 0.01$ | $-0.15 \pm 0.03$ |
| $\frac{\partial F}{\partial S}\Delta S$ | $-0.01 \pm 0.0$ | $0.01 \pm 0.01$ | $0.0 \pm 0.0$ | $-0.0 \pm 0.0$ | $-0.02 \pm 0.0$ |
| $\frac{S}{S_0}\frac{\partial F}{\partial DIC}\Delta sDIC$ | $0.11 \pm 0.04$ | $-0.34 \pm 0.05$ | $-0.04 \pm 0.01$ | $-0.8 \pm 0.05$ | $0.33 \pm 0.05$ |
| $\frac{S}{S_0}\frac{\partial F}{\partial Alk}\Delta sAlk$ | $-0.01 \pm 0.02$ | $0.06 \pm 0.02$ | $0.01 \pm 0.0$ | $0.07 \pm 0.01$ | $-0.01 \pm 0.02$ |
| $\frac{\partial F}{\partial fw}\Delta fw$ | $-0.01 \pm 0.0$ | $0.01 \pm 0.01$ | $0.0 \pm 0.0$ | $0.0 \pm 0.0$ | $-0.02 \pm 0.0$ |
| *Sum of Terms Versus Modeled* | | | | | |
| $\Sigma$ | $-0.03 \pm 0.01$ | $0.02 \pm 0.02$ | $-0.01 \pm 0.0$ | $-0.38 \pm 0.05$ | $0.21 \pm 0.03$ |
| $\Delta F$ | $-0.10 \pm 0.01$ | $0.12 \pm 0.02$ | $-0.04 \pm 0.01$ | $-0.49 \pm 0.06$ | $0.2 \pm 0.02$ |

[1] $\mathrm{mol\ m^{-2}\ yr^{-1}\ \sigma^{-1}}$
[2] $\mathrm{mol\ m^{-2}\ yr^{-1}\ K^{-1}}$

**Table 3.** Regression coefficients between the given EBUS and climate index for anomaly time series of the estimated contributions toward the sDIC tendency integrated over the upper 100m and the total surface area of the system, as in Equation 5.[1]

| Term | CalCS – NPGO[1] | HumCS – Nino3[2] | CanCS – NAO[1] |
|---|---|---|---|
| $\frac{dsDIC'}{dt}$ | $0.03 \pm 0.01$ | $-0.10 \pm 0.02$ | $0.04 \pm 0.01$ |
| $J'_{ex}$ | $-0.76 \pm 0.20$ | $-4.84 \pm 0.61$ | $0.51 \pm 0.21$ |
| $J'_{bio}$ | $-0.68 \pm 0.48$ | $8.61 \pm 0.81$ | $-0.49 \pm 0.54$ |
| $J'_{circ}$ | $1.47 \pm 0.64$ | $-3.87 \pm 1.28$ | $0.02 \pm 0.55$ |

[1] TgC yr$^{-1}$ $\sigma^{-1}$
[2] TgC yr$^{-1}$ K$^{-1}$